# Toxicopathological studies on the effects of T-2 mycotoxin and their interaction in juvenile goats

**Shivasharanappa Nayakwadi**[1,2]\*, **Ramith Ramu**[3], **Anil Kumar Sharma**[1,4], **Vivek Kumar Gupta**[5], **K. Rajukumar**[6], **Vijay Kumar**[1], **Prithvi S. Shirahatti**[7], **Rashmi L.**[8], **Kanthesh M. Basalingappa**[9]

**1** Central Institute for Research on Goats (CIRG), Makhdoom, Mathura, India, **2** Animal Science Section, ICAR-Central Coastal Agricultural Research Institute, Ela, Goa, India, **3** Division of Biotechnology and Bioinformatics, Department of Water & Health Sciences–Faculty of Life Sciences, JSS Academy of Higher Education and Research (Deemed to be University), Mysuru, India, **4** Division of Pathology, Mycotic and Mycotoxic Diseases Laboratory, Indian Veterinary Research Institute, Izatnagar, India, **5** CADRAD, ICAR–Indian Veterinary Research Institute, Izatnagar, India, **6** ICAR–National Institute of High Security Animal Diseases, Bhopal, India, **7** Department of Biotechnology, Teresian College, Siddhartha Nagara, Mysuru, India, **8** Karnataka Veterinary Animal Fisheries University, Bidar, Karnataka, India, **9** Division of Molecular Biology, Department of Water & Health Sciences–Faculty of Life Sciences, JSS Academy of Higher Education and Research (Deemed to be University), Mysuru, India

\* drshivasharan@gmail.com, shivasharanappa.n@icar.gov.in

**Data Availability Statement:** All relevant data are within the paper.

## Abstract

Food and feeds contaminated with mycotoxins have been a threat to the rearing industry by causing some of the most fatal toxic reactions not only in the farm animals but also in humans who consume them. Toxicity to juvenile goats was induced by feed contamination with T-2 toxin (at 10 and 20 ppm dosage; group I and II, respectively). The toxicity impact was assessed on days 15 and 30 post treatment with respect to growth performance, oxidative stress, apoptotic studies and detailed pathomorphology. The study revealed that apart from the obvious clinical toxicosis (weakness, lethargy, and retardation in growth), the toxin fed groups also exhibited significant haematological (reduced hemoglobin, total leukocyte and thrombocyte counts) and biochemical changes (increased levels of oxidative stress markers with concomitant decrease in levels of serum and tissue catalase and superoxide dismutase). The pathomorphological and histological alterations suggested that the liver and intestine were the most affected organs. Ultra-structurally, varying degrees of degeneration, cytoplasmic vacuolations and pleomorphic mitochondria were observed in the hepatocytes and the enterocytes of the intestine. Kidney also revealed extensive degeneration of the cytoplasmic organelles with similar condensation of the heterochromatin whereas the neuronal degeneration was characterized by circular, whirling structures. In addition, the central vein and portal triad of the hepatocytes, cryptic epithelial cells of the intestine, MLNs in the lymphoid follicles, PCT and DCT of the nephronal tissues and the white pulp of the spleen exhibited extensive apoptosis. In this study, it was also observed that the expression of HSPs, pro-apoptotic proteins and pro-inflammatory cytokines were significantly upregulated in response to the toxin treatment. These results suggest that the pathogenesis of T-2 toxicosis in goats employs oxidative, apoptotic and inflammatory mechanisms.

**Funding:** The entire work was not funded by any agency.

**Competing interests:** The authors have declared that no competing interests exist.

## Introduction

Mycotoxins *viz.*,aflatoxins, zearelenone, ochratoxins, fumonisins, citrinins and trichothecenes are the secondary metabolites produced by fungi responsible for mycotoxicosis in animals. Recent studies have reported the consequences of food contamination with mycotoxins, which are known to pose a major problem not only in animals but also to humans, causing significant economic and international trade implications [1]. As a result of the mycotoxin contamination in animal feed, there is reduced feed intake and at times feed refusal, reduced body weight due to inefficient of feed conversions, immune- suppression leading to frequent diseases, reduced reproductive ability [2–5]. Trichothecenes are a class of mycotoxins that are identified as a major threat to animals and healthcare. Most toxic trichothecenes are deoxynivalenol (DON), nivalenol (NIV), HT-2, diacetoxyscirpenol (DAS) and T-2 toxin[6]. Of these, T-2 toxin, mainly produced by *Fusarium* species, is the most deadly toxin with cytotoxic potential. This toxin has been considered to elicit fatal reactions among the animals and humans upon consumption [7]. The production of toxins by the fungi occur at ambient temperature of 0–32˚C and moisture content [8,9]. Frequently contaminated grains are corn, wheat, rye, oats and barley. Studies have shown that T-2 toxicity leads to reduced feed consumption and milk production, absence of estrus cycle, gastroenteritis, intestinal hemorrhages and necrosis in cattle [10–17],. The T-2 toxicity has also been well studied in laboratory animals such as sheep[18,19], pigs[20] and poultry[21]. However, toxicity in goat husbandry is not much explored.

It is observed that the presence of antioxidant scavengers increase the toxicity of the mycotoxins, suggesting that the toxicity is probably augmented by the presence of oxidative stress. Heat shock proteins (HSPs) play a pivotal role in rendering a protective shield towards oxidative damage induced by the mycotoxins. Studies have confirmed the induction of high ROS generation by T-2 toxin, which in turn increases the levels of Hsp70. Therefore, the existence of an intricate relationship between oxidative stress and HSPs can be used as a potent biomarker for assessing oxidative stress [22–24]. Oxidative stress decides the apoptotic fate of the cell by eliciting a cascade of pro-apoptotic pathways. In addition, T-2 toxins are known to induce the expression of various pro-inflammatory cytokines following a tropical application in rats [25] and increased the expression of IL-12 and TNF-α with a corresponding decrease in the expression of IL-1β in mouse peritoneal macrophages and lymph node T cells [26]. In addition, in previous findings this toxin has been observed to increase the expression of IL-2 and reduce IFN-γ expression in pigs [27]. T-2 toxin is known to induce apoptosis in lymphoid and hematopoietic tissues, intestinal crypt epithelial cells, spleen as well as the brain [28–31].

With this background, the present study is to ascertain T-2 toxin-induced pathology in goats. In addition, the study aims at the assessment of expression of various HSPs as oxidative damage indicators, pro-apoptotic genes and pro-inflammatory cytokines in goats in order to understand the molecular mechanism of T-2 toxicity.

## Materials and methods

### Production and analysis of T-2 toxin

Pure culture of *Fusarium sporotrichioides var. sporotrichioides* (MTCC-1894) was procured from the Institute of Microbial Technology (IMTC), Chandigarh, India. T-2 mycotoxin was produced by fermentation of maize and wheat mixture as per the method described in AOAC international [32]. The generated toxin was analyzed and quantified by thin layer chromatography and spectrophotometry at the Animal Feed Analytical and Quality Control Laboratory (AFAQCL), Veterinary College and Research Institute, Namakkal, Tamil Nadu, India.

## Experimental animals

A total of eighteen goats (Barbari breed) between 2–3 months age were used in this study. All the procedures in the trial were carried out under strict guidelines with the recommendations and prior approval by the Institute's Animal Ethics committee (IAEC) of ICAR-Central Institute for Research on Goats (CIRG), Mathura and Committee for the Purpose of Control and Supervision of Experiments on Animals (CPCSEA), New Delhi (India) (number: 207\GO/cb/2000/CPCSEA dt:1.6.2000). The animals were examined and treated for internal and external parasites and maintained under standard conditions. After an acclimatization period of one week, the toxin was administered orally by mixing with the feed along with a control group. These animals were kept under optimum housing conditions and provided ad libitum feed and fresh drinking water. After the trial, the goats were euthanized by intravenous administration of 20% sodium pentobarbital into the jugular vein with minimal pain and suffering. During the course of the experiment, no animals died prior to euthanization [33].

## Experimental diet

Feed pellets as well as ground feed were procured from the Feed Processing Unit, C.I.R.G, Makhdoom and were tested negative for the presence of mycotoxin contamination (Aflatoxin $B_1$, Ochratoxin A, citrinin and T-2 toxin) The inoculum containing known amounts of T-2 toxin was added and thoroughly mixed with the basal ration in proportion to deliver the set dosage (10 and 20 ppm). From the diet, aliquots were taken and quantified by TLC and spectrophotometric analysis for assuring the correct concentration.

## Experimental design

Juvenile goats were randomly divided into three groups consisting of six animals each and weighed to give an average initial mean body weight and tags were marked for identification. Feed containing T-2 toxin levels of 10 ppm and 20 ppm (mg/kg of feed) were prepared. The details of the experimental design are given in (Table 1).

## Parameters studied

**a. Clinical signs.** Careful observation of the animals in the study was conducted to evaluate the development of any clinical conditions or mortality. Body weight of individual animal was recorded on day 0, 5, 10, 15 and 20, 25 and 30 post-treatment in all the groups.

**b. Hematology.** Blood was drawn from the jugular vein from all the goats of different groups on day 0, 5, 10, 15 and 20, 25 and 30 in vials containing Ethylene diamine tetra acetate (1mg/ml of blood). Hemoglobin concentration (Hb), packed cell volume (PCV), total erythrocyte count (TEC), total leukocyte counts (TLC), differential leukocyte counts (DLC) and total

**Table 1. Experimental design of T-2 toxicosis in goats.**

| Details | Group I (10 ppm) | Group II (20 ppm) | Group III (control) |
|---|---|---|---|
| Age of goats (kids) | 2–3 months | 2–3 months | 2–3 months |
| No. of animals | 06 | 06 | 06 |
| Duration of the experiment | 30 days | 30 days | 30 days |
| Dose of toxin | 10 ppm (mg/kg of feed) | 20 ppm (mg/kg of feed) | Toxin free feed |
| Route of administration | Oral (Mixed with feed) | Oral (Mixed with feed) | Oral (Feed free from mycotoxins) |
| Sacrifice intervals | 15th and 30th day of the trial | | |
| No. of animals sacrificed at each interval | 03 | 03 | 03 |

thrombocyte count (TTC) were carried out by automatic blood analyzer(Sysmex XE-5000 hematology analyzer, Sysmex, Kobe, Japan) as per the manufacturer's instructions.

**c. Estimation of oxidative stress enzymes.** The method to evaluate oxidative stress were TBRS activity (Thiobarbituric acid 'TBA' method), Superoxide dismutase (SOD) and catalase in liver, kidney, small intestine, spleen, mesenteric lymph node (MLN) and brain tissue homogenates for all the animals on 15th and 30th day according to the method described by Shivasharanappa [34]

## Pathomorphological studies

**Gross pathology.** Post euthanization procedure on the 15th and 30th day, the visceral organs was carefully evaluated for the development of lesions. The weights of these organs were also tabulated as relative organ weight.

**Histopathology.** Pieces of the tissues from liver, intestines, MLN, kidneys, spleen and brain (less than 5 mm thickness) were collected and fixed in 10% neutral buffered formalin solution. After 48 hours of fixation, the tissues were processed and stained by hematoxylin and eosin method [35].

**Ultra-structural changes.** Approximately 1-2mm sized tissue pieces from liver, kidney, small intestine and brain were collected and fixed in chilled 2.5% glutaraldehyde for 6–8 h at 4° C. Tissue samples were washed twice (20 min each) in chilled 0.1 M phosphate buffer and then fixed in 1% osmium tetroxide for 2 h at 4° C. The tissues were rinsed thrice in 0.1 M phosphate buffer at 4° C and dehydrated in ascending grades of acetone (30%, 50%, 70%, 90%, 100% and dry acetone) for 15 min each. The tissue samples were cleared twice in toluene at room temperature and infiltrated in toluene and resin in the ratio 3:1, 2:2, 1:3 and pure resin for one hour. The tissue samples were embedded in Araldehyte media and kept at 37°C for 24 h and then incubated at 45° C for 2 days for polymerization. The polymerized blocks were trimmed and ultra-sections of 500 nm were cut employing ultra tome microscope (Ultra-cut UCT, Leica, Germany), mounted on copper grids and stained with uranyl acetate [36] and followed by lead citrate[37].The sections were washed and allowed to dry on a filter paper in a covered Petri dish. The grids were mounted to be viewed under the electron microscope (Philips, CM-10, Holland). Processing, section cutting, staining and examination of grids were carried out at High Security Animal Disease Laboratory (HSADL), Bhopal. India.

## Apoptosis by TUNEL (TdT-mediated dUTP Nick-End Labeling) assay

Apoptosis was studied using Dead End™ Colorimetric TUNEL System (Promega) assay in liver, intestines, kidney, spleen, mesenteric LN and brain. After the staining, the TUNEL-positive cells were counted under the scanning confocal microscope to evaluate apoptosis of the cells.

**RNA isolation and cDNA synthesis.** The apoptosis genes such as Bcl-2, Bax and caspase-3, Heat shock protein (HSP) genes viz. HSP-72, 90 and 27 and pro-inflammatory genes (IL-1, IL-6, TNF-$\alpha$) in different tissues such as liver, intestine, kidney, spleen, mesenteric LN and brain were used to isolate RNA. RNeasy Lipid Mini Kit (Qiagen, Hilden, Germany) was used to extract total RNA from the tissue samples. The concentration and quality was evaluated using Nano Drop 2000 spectrophotometer (Thermo Fischer Scientific Inc, Carlsbad, CA, USA). Complementary DNA (cDNA) synthesis was performed using the Superscript First Strand Synthesis System for RT-PCR (Invitrogen, Carlsbad, CA, USA). PCR amplification was carried out using the cycling conditions as follows: an initial 10 min activation and denaturation step at 95°C, followed by 40 cycles of 15s at 95°C and 30 s, 72°C for 1 min and finally 72°C for 5 min.

**Differential expression of HSPs (HSP-72, 90 and 27), apoptosis (Bcl-2, Bax and Caspase 3) genes and pro-inflammatory cytokines (IL-1, IL-6 and TNF-α) using qRT-PCR.** The reaction mixture containing 1X SYBR®Green (PE Applied Biosystems), 5mM $MgCl_2$, buffer, 0.5μM primer, 1 U Taq, 0.3 mM dNTP and 50 ng cDNA was subjected to the PCR reaction using the cycler conditions mentioned above. Real-time RT-PCR amplifications were performed in an ABI-Prism 7000 Sequence Detection System (PE Applied Biosystems). The levels of gene expression were determined using the comparative Ct method. A normalization factor was used to determine the expression level of each gene in each sample against the control. All target gene transcriptions were expressed as n-fold difference relative to the control.

## Statistical analysis

The data was statistically analyzed using analysis of variance (ANOVA) and significance levels were kept at 1 and 0.5 percent. For the analysis of qPCR products, the data obtained were analyzed by using the $2^{-(\Delta\Delta Ct)}$ method. Finally, the mean relative expression for each group was statistically analyzed by one way ANOVA [least significance difference (LSD) and Duncan's Test] for 'T' distribution (P value) using SPSS (21.0) Software.

## Results

### Clinical signs

The administration of the toxin to goats exhibited varying degree of pathological symptoms including weakness, lethargy, growth retardation, feed refusal and reduced movement. However, diarrhea was the most consistent symptom observed right from the 20th day post treatment in both the toxin treated groups. 4 goats from group I and 6 from group II showed diarrhea with soiled hindquarters. Goats from the group III remained void of these clinical symptoms. Also, in group II extensive pathological lesions were seen in the lumen and abomasum of the GI system. Further, ruminal epithelium exhibited peeling whereas abomasum showed congestion in response to the toxin treatment on 15th and 30th day. However, no significant histological changes were seen in group I on 15th and 30th day.

### Body weight

The body weights recorded at 0, 5, 10, 15, 20, 25 and 30th day post feeding of the toxin are given in (Table 2). 15 days onwards, the mean body weights in the toxin fed groups exhibited lower weight-gain than in the control group, which were found to be dose- and duration-dependent. On 15th and 30th day, group II showed significantly lower body weight-gain when compared to other groups. In the control group, there was a progressive body weight-gain with respect to time contrary to those in the toxin fed groups. No mortality was observed during the experimental period.

### Haematological changes

The hematological parameters are tabulated in (Table 3). There was a significant reduction in hemoglobin, total leukocyte counts and total thrombocyte counts in toxin treated groups in comparison with those of the control group in a dose- and time-dependent manner. There was a significant reduction in total platelet count particularly in group II on the 25th day.

### Oxidative stress enzymes

The antioxidant enzyme values are tabulated in (Table 4). The antioxidant enzymes were remarkably enhanced in the tissue homogenates around the 30th day suggesting its time

**Table 2. Effect of T-2 toxin on body weights of kids fed with 10 ppm and 20 ppm dose at different intervals.**

| Days | Body weight (kg) | | |
|---|---|---|---|
| | Group I (10ppm) | Group II (20ppm) | Group III (Control) |
| 0 | 7.05±0.34[aA] | 7.08±0.86[aA] | 6.79±0.49[aA] |
| 5 | 7.09±0.42 [abA] | 6.79±0.44 [aA] | 6.62±0.27 [bA] |
| 10 | 7.43±0.28 [aA] | 6.29±0.47 [aB] | 7.47±0.19 [aA] |
| 15 | 7.16±0.42 [abA] | 5.92±0.48 [aB] | 7.25±0.19 [abA] |
| 20 | 6.280±0.1 [cbaA] | 5.89±0.62 [aA] | 7.36±0.12 [aA] |
| 25 | 6.02±0.39 [cbAB] | 5.56±0.52 [aB] | 7.73±0.06 [aA] |
| 30 | 5.65±0.27 [cB] | 5.21±0.52 [aB] | 7.73±0.1 [aA] |

Means bearing at least one common superscript (a,b,c) do not differ significantly between groups (p<0.05). Means bearing at least one common superscript (A, B) do not differ significantly between days (p<0.05). $F_D$ = 1.49, $F_G$ = 14.91**, $F_{GD}$ = 1.89, $F_D$ = effect within days (D), $F_G$ = effect within groups (G), $F_{DG}$ = interaction within days and groups *<0.05, **<0.01

dependency. Hepatic SOD and catalase were notably increased in group II than in group I whereas hepatic MDA levels were higher in group I even from the 15th day onwards indicating its toxicity on the liver at lower doses also. Likewise, the intestinal SOD and catalase levels were higher in group II on 3th day indicating a severe injury to the cells of the intestine. In addition, in the MLN and spleen the MDA concentration increased remarkably whereas only catalase and SOD increased in the spleen in group II. Also, in the brain levels of all the enzymes tested were increased with no notable difference between the two groups.

## Pathomorphological changes

**Gross pathological alterations.** *Group I.* Overall, remarkable changes were found in most visceral organs except in intestine, liver and spleen. The relative organ weights of the liver and MLN increased. There was a marked enlargement of the liver and diffused congestion with distended gall bladder on the 30th day of the trial. All the segments of the intestine showed diffused congestion and haemorrhages with pasty mucoid contents. Kidneys at 15th and 30th day showed paler cortex. The other organs such as abomasum, rumen, lungs and heart did not exhibit any remarkable gross changes.

**Table 3. Effect of T-2 toxin on hemoglobin (Hb), total leukocyte counts (TLC) and total thrombocyte count (TTC) at different intervals in goats fed with 10 ppm and 20 ppm of toxin.**

| Parameters | GROUP | 5th | 10th | 15th | 20th | 25th | 30th | F value |
|---|---|---|---|---|---|---|---|---|
| Hb (g/dl) | I | 07.62±0.37[bc] | 08.06±0.65[c] | 07.87±0.29[c] | 07.73±0.68[bc] | 5.32±0.14[abB] | 5.01±0.03[aB] | $F_D$ = 23.92** |
| | II | 07.43±0.29[b] | 07.40±0.21[b] | 07.50±0.45[b] | 06.33±0.24[b] | 4.21±0.03[aA] | 4.06±0.02[aA] | $F_G$ = 5.34** |
| | Control | 07.65±0.24[a] | 07.48±0.23[a] | 07.20±0.21[a] | 07.07±0.18[a] | 7.01±0.04[aC] | 6.98±0.08[aC] | $F_{DG}$ = 4.61** |
| TLC (x10³/µl) | I | 07.00±0.29 | 08.82±0.56[A] | 06.80±0.98 | 07.51±0.23[B] | 5.56±0.23[aAB] | 4.17±0.32[aA] | $F_D$ = 10.68** |
| | II | 07.98±0.47[c] | 06.43±0.18[bcB] | 07.43±0.33[c] | 04.59±0.19[abA] | 4.53±0.61[abA] | 3.67±0.18[aA] | $F_G$ = 8.47** |
| | Control | 06.75±0.52[a] | 06.50±0.28[aB] | 06.03±0.08[a] | 07.93±0.57[bB] | 6.98±0.86[abB] | 7.05±0.09[abB] | $F_{DG}$ = 11.7** |
| TTC (x10³/µl) | I | 357.30±38.5[a] | 402.80±31.[6a] | 344.20±24.3[a] | 309.20±7.19[aB] | 265.50±6.54[aB] | 253.3±5.43[aB] | $F_D$ = 5.67** |
| | II | 371.20±44.8[ab] | 415.90±29.0[b] | 368.40±54.1[ab] | 225.30±11.7[abA] | 199.10±3.23[aA] | 176.30±6.65[aA] | $F_G$ = 11.43** |
| | Control | 399.1±39.3[a] | 382.70±33.3[a] | 476.30±5.9[a] | 396.50±.22[aC] | 383.0±6.65[aC] | 392.50±4.56[aC] | $F_{DG}$ = 1.7 |

The data were analyzed using two way-ANOVA and Tukey's Post hoc test which is incorporated in SPSS statistical software. Superscripts A, B and C shows differences in columns and a, b, c, d in rows. Mean bearing at least one common superscript didn't show any difference. FD = effect within days (D), FG = effect within groups (G), FDG = interaction within days and groups *<0.05, **<0.01.

**Table 4. Effect of T-2 toxin on oxidative stress enzymes in various organs.**

| Organ | Enzymes | (Days) | Groups | | | F value |
|---|---|---|---|---|---|---|
| | | | 10ppm (Group I) | 20ppm (Group II) | Control (III) | |
| Liver | CAT ($10^3$/U) | 15 | $1756.50\pm3.15^{aB}$ | $1684.2\pm9.45^{bB}$ | $1629.2\pm1.95^{cB}$ | FG = 226.4** |
| | | | | | | FD = 305.7** |
| | | 30 | $1843.90\pm5.70^{bA}$ | $1939.4\pm17.8^{aA}$ | $1659.3\pm1.9^{cA}$ | FDG = 88.85** |
| | SOD (U/g) | 15 | $77.10\pm1.25^{bB}$ | $86.6\pm2.30^{aB}$ | $73.2\pm0.55^{bB}$ | FG = 72.32** |
| | | 30 | $106.60\pm4.05^{aA}$ | $118.8\pm1.6^{aA}$ | $82.9\pm0.80^{bA}$ | FD = 187.0** |
| | | | | | | FDG = 17.0** |
| | LPO (nM) | 15 | $56.00\pm2.65^{aB}$ | $63.4\pm4.15^{a}$ | $33.9\pm1.4^{b}$ | FG = 90.9** |
| | | 30 | $78.80\pm4.40^{aA}$ | $80.9\pm4.15^{a}$ | $29.8\pm0.5^{b}$ | FD = 20.2** |
| | | | | | | FDG = 9.45* |
| Intestine | CAT ($10^3$/U) | 15 | $1578.30\pm4.95^{bB}$ | $1804.6\pm5.8^{aB}$ | $1436.3\pm2.4^{cB}$ | FG = 566.0** |
| | | 30 | $2037.70\pm6.5^{aA}$ | $1949.8\pm24.8^{bA}$ | $1619.3\pm4^{cA}$ | FD = 854.9** |
| | | | | | | FDG = 121.8** |
| | SOD (U/g) | 15 | $85.90\pm0.35^{bB}$ | $110.8\pm1.55^{aB}$ | $76.5\pm0.45^{c}$ | FG = 422.6** |
| | | 30 | $106.30\pm2.05^{bA}$ | $134.4\pm2.15^{aA}$ | $78.6\pm1.65^{c}$ | FD = 123.5** |
| | | | | | | FDG = 30.9** |
| | LPO (nM) | 15 | $57.40\pm3.10^{bB}$ | $96.3\pm3.6^{a}$ | $48.1\pm2.55^{b}$ | FG = 206.5** |
| | | 30 | $98.40\pm2.0^{aA}$ | $113.7\pm4.0^{a}$ | $41.7\pm1.9^{b}$ | FD = 51.0** |
| | | | | | | FDG = 32.1** |
| MLN | CAT ($10^3$/U) | 15 | $1602.80\pm9.4^{bB}$ | $1649\pm4.65^{aB}$ | $1366.1\pm3.4^{cB}$ | FG = 1009.6** |
| | | 30 | $1665.20\pm8.7^{bA}$ | $1751.8\pm6.05^{aA}$ | $1482.4\pm3.2^{cA}$ | FD = 331.1** |
| | | | | | | FDG = 10.5* |
| | SOD (U/g) | 15 | $74.80\pm2.45^{aB}$ | $78.6\pm0.85^{aA}$ | $45.6\pm2.15^{b}$ | FG = 266.5** |
| | | 30 | $86.0\pm1.55^{aA}$ | $70.5\pm1.05^{bB}$ | $52.2\pm0.9^{c}$ | FD = 11.3* |
| | | | | | | FDG = 24.9** |
| | LPO (nM) | 15 | $56.4\pm1.45^{B}$ | $58.6\pm1.05^{B}$ | $50.9\pm3.7$ | FG = 30.3** |
| | | 30 | $70.4\pm2.45^{aA}$ | $79.4\pm3.45^{aA}$ | $50.3\pm1.35^{b}$ | FD = 35.0** |
| | | | | | | FDG = 8.7* |
| Kidneys | CAT ($10^3$/U) | 15 | $1717.60\pm15.20^{b}$ | $1789.4\pm7.2^{aA}$ | $1571.6\pm2.4^{bA}$ | FG = 78.8** |
| | | 30 | $1717.00\pm6.40^{a}$ | $1651.5\pm3.8^{bB}$ | $1590.6\pm1.35^{cB}$ | FD = 150.7** |
| | | | | | | FDG = 41.9** |
| | SOD (U/g) | 15 | $74.80\pm2.45^{b}$ | $116.4\pm3.00^{a}$ | $71.2\pm1.1^{c}$ | FG = 292.6** |
| | | 30 | $84.60\pm2.25^{b}$ | $135.3\pm3.5^{a}$ | $66.5\pm2.1^{c}$ | FD = 15.6** |
| | | | | | | FDG = 11.7** |
| | LPO (nM) | 15 | $60.50\pm3.9$ | $61.4\pm2.5$ | $49.6\pm4.35$ | FG = 16.95** |
| | | 30 | $73.50\pm2.55$ | $61.9\pm4.35$ | $48.1\pm6.2$ | FD = 0.8 |
| | | | | | | FDG = 3.88 |
| Spleen | CAT ($10^3$/U) | 15 | $1431.20\pm1.65^{cB}$ | $1766.3\pm4.45^{aB}$ | $1573.8\pm2.5^{bB}$ | FG = 1904.4** |
| | | | | | | FD = 944.3** |
| | | 30 | $1642.80\pm3.55^{bA}$ | $1829.3\pm8.55^{aA}$ | $1630.2\pm1.4^{bA}$ | FDG = 197.6** |
| | SOD (U/g) | 15 | $67.30\pm0.95^{bB}$ | $81.3\pm1.95^{a}$ | $56.9\pm2.6^{b}$ | FG = 78.7** |
| | | 30 | $73.40\pm1.00^{bA}$ | $89.3\pm0.95^{a}$ | $65.3\pm3.05^{b}$ | FD = 22.5** |
| | | | | | | FDG = 0.22 |
| | LPO (nM) | 15 | $70.40\pm2.75^{b}$ | $91.4\pm1.75^{aB}$ | $74.9\pm2.75^{b}$ | FG = 65.99** |
| | | 30 | $74.40\pm4.80^{b}$ | $122.2\pm3.8^{aA}$ | $81.25\pm2.05^{b}$ | FD = 27.5** |
| | | | | | | FDG = 10.6* |

(*Continued*)

**Table 4.** (Continued)

| Organ | Enzymes | (Days) | Groups | | | F value |
|---|---|---|---|---|---|---|
| | | | 10ppm (Group I) | 20ppm (Group II) | Control (III) | |
| Brain | CAT ($10^3$/U) | 15 | 1626.60±12.90$^{bB}$ | 1717.7±5.6$^{aB}$ | 1564.9±2.75$^{cA}$ | FG = 1675.2** |
| | | | | | | FD = 9.84* |
| | | 30 | 1717.50±5.80$^{bA}$ | 1909.7±1.4$^{aA}$ | 1329.2±1.6$^{cB}$ | FDG = 610** |
| | SOD (U/g) | 15 | 72.40±0.75$^{bB}$ | 94.2±1.00$^{a}$ | 65.1±2.2$^{b}$ | FG = 115.9** |
| | | 30 | 82.20±0.70$^{abA}$ | 88.30±1.05$^{a}$ | 74.2±2.1$^{b}$ | FD = 14.2** |
| | | | | | | FDG = 19.23* |
| | LPO (nM) | 15 | 79.90±3.25$^{b}$ | 99.40±1.65$^{a}$ | 77.9±3.8$^{b}$ | FG = 33.9** |
| | | 30 | 82.40±5.40$^{b}$ | 108.30±2.25$^{a}$ | 79.3±2.45$^{b}$ | FD = 2.7 |
| | | | | | | FDG = 0.81 |

The data were analysed using one way ANOVA, two way ANOVA and independent t test wherever applicable. Superscripts A, B and C shows differences in columns and a, b, c, d in rows. Mean bearing at least one common superscript didn't show any difference. $F_D$ = effect within days (D), $F_G$ = effect within groups (G), $F_{DG}$ = interaction within days and groups *<0.05, **<0.01. CAT ($10^3$/U)—Catalase, SOD-Superoxide Dismutase, LPO-lipid peroxidation (nM of MDA/g of tissue).

*Group II.* At the 30$^{th}$ day post toxin exposure, liver was found enlarged with rounded edges, diffused congestion and distended gall bladder. The intestine and liver were the most affected organs in this group. Mesenteric lymph nodes were enlarged and oedematous and the brain revealed meningeal vessels engorgement.

## Histological alterations

Extent, severity and type of lesions differed among the groups based on the type of organ involved, dose and duration of toxin feeding. Liver and intestines were the most severely affected organs exhibiting remarkable histological alterations. These lesions were broadly classified as vascular and cellular changes in all the organs. Overall scoring of histological lesions was also done for the other organs mentioned above. The histopathological changes observed in various organs of the different groups are given in (Table 5).

**Liver.** *Group I.* Varying degree of degenerative and necrotic lesions were seen in the hepatic parenchyma. Hepatocytes on the 15$^{th}$ day exhibited swollen and granular cytoplasm. Bi-nucleated hepatocytes having karyomegaly were observed at various instances whereas a few cells also exhibited degeneration and necrosis of the eosinophilic granular cytoplasm and nuclear condensation (**Fig 1**).

30$^{th}$ day post toxin treatment showed extensive lesions characterized by degeneration of the hepatocytes towards the portal triad. Hepatic artery and portal veins along with the other blood vessels were swollen and dilated. Hyperplasia of the bile duct was also visible along with cell necrosis consistently present on the 30$^{th}$ day. In one case, diffused type of coagulative necrosis of hepatocytes (individual cell necrosis) with dark pyknotic nucleus was seen scattered in the parenchyma along with diffuse fatty changes in large number of hepatocytes (**Fig 2**).

*Group II.* This group exhibited the maximum severity of lesions of the hepatocytes right from the 15$^{th}$ day showing engorgement and sinusoidal congestion in all the experimental animals. Visible centrilobular necrosis was found along with hypertrophy of the bile duct and dead cell debris accumulation in the lumen leading to peri-billiary fibrosis (**Fig 3**).

By the 30$^{th}$ day, all these features were observed in all the animals in the study. Extensive cell necrosis was seen with distinct and dark nucleus as well as eosinophilic cytoplasm called 'Ghost hepatocytes' (**Fig 4**).

**Intestines.** *Group I.* The segments of the intestine namely duodenum, ilium and jejunum showed characteristic lesions led by cellular and vascular alteration by 15$^{th}$ day post toxin

**Table 5. T-2 toxin induced histological alterations (score card) in different organs of goats on 15th and 30th days fed with 10ppm and 20ppm dose.**

| Organs | Lesions | 15th day | | | At 30th day | | |
| --- | --- | --- | --- | --- | --- | --- | --- |
| | | Groups | | | Groups | | |
| | | I | II | III | I | II | III |
| **Liver** | Vascular engorgement/sinusoidal congestion | - | 1M/3 | 1m/3 | 1m/3 | 3M/3 | 1m/3 |
| | Hepatocyte degeneration/coagulative necrosis | 1M/3 | 2M/3 | 0/3 | 3M/3 | 3S/3 | 1m/3 |
| | Centrio-lobular degeneration/ coagulative necrosis | 1m/3 | 3S/3 | 0/3 | 2S/3 | 3S/3 | 0/3 |
| | Hepatocyte vacuolation | 1M/3 | 1M/3 | 0/3 | 1m/3 | 1S/3 | 0/3 |
| | Bile duct hyperplasia/thickening | - | 3S/3 | 0/3 | 2S/3 | 3S/3 | 0/3 |
| | Periductular connective tissue proliferation/thickening/fibrosis | - | 2S/3 | 0/3 | - | 3S/3 | 0/3 |
| | Total score | 05 | 32 | 01 | 20 | 48 | 02 |
| **Intestine** | Vascular engorgement/congestion | 1M/3 | 2M/3 | 1m/3 | 1m/3 | 3S/3 | 0/3 |
| | Crypt epithelial hyperplasia | 1M/3 | 2M/3 | 1m/3 | 3S/3 | 3S/3 | 1m/3 |
| | Necrosis of cryptic cells | - | 2S/3 | 0/3 | 3S/3 | 3S/3 | 0/3 |
| | Infiltration of lymphocytes in lamina propria | - | 2m/3 | 1m/3 | 3M/3 | 3S/3 | 1M/3 |
| | Necrosis of lymphocytes in lamina propria | - | 2S/3 | 0/3 | 2S/3 | 3S/3 | 0/3 |
| | Giant crypts/fusion of crypts | - | 1m/3 | 0/3 | - | 3M/3 | 0/3 |
| | Total score | 04 | 23 | 03 | 31 | 51 | 03 |
| **MLN** | Vascular engorgement | - | - | 0/3 | - | 1m/3 | 0/3 |
| | Lymphoid depletion | - | 1m/3 | 0/3 | 1S/3 | 2M/3 | 0/3 |
| | Lymphocyte necrosis/nuclear fragmentation/apoptosis | 1m/3 | 2S/3 | 0/3 | 3S/3 | 3S/3 | 0/3 |
| | Total score | 01 | 07 | 0 | 12 | 14 | 0 |
| **Kidneys** | Vascular congestion/hemorrhages | 1m/3 | 2m/3 | 0/3 | 2m/3 | - | 0/3 |
| | Interstitial edema/plasma leakage | - | 1m/3 | 0/3 | 1m/3 | 1m/3 | 0/3 |
| | Tubular degeneration/necrosis | 1M/3 | 3S/3 | 0/3 | 3S/3 | 3S/3 | 0/3 |
| | Total score | 03 | 12 | 0 | 12 | 10 | 0 |
| **Spleen** | Vascular engorgement | 1m/3 | 2m/3 | 0/3 | 2m/3 | 1S/3 | 0/3 |
| | Lymphoid depletion | - | 2M/3 | 0/3 | 2S/3 | 2S/3 | 0/3 |
| | Lymphocyte necrosis/apoptosis | - | 2M/3 | 0/3 | 2S/3 | 3S/3 | 0/3 |
| | Epitheloid cell reaction/Macrophage infiltration | - | 1M/3 | 0/3 | - | 3S/3 | 0/3 |
| | Total score | 01 | 12 | 0 | 20 | 27 | 0 |
| **Brain** | Vascular engorgement | - | - | 0/3 | 2M/3 | 2M/3 | 1m/3 |
| | Neuronal degeneration | - | 1m/3 | 0/3 | 1M/3 | 2M/3 | 1m/3 |
| | Focal/diffuse gliosis | - | 1m/3 | 0/3 | 2M/3 | 1M/3 | 0/3 |
| | Total score | - | 02 | 0 | 08 | 08 | 02 |

Group I-10 ppm, II-20 ppm and III- Control; Scoring–Number of organs with lesions/Number of organs examined per group; m- mild, M-moderate, S-severe. Score: m = 01, M = 02, S = 03.

induction. Hyperplasia of the crypt epithelium was moderate whereas less evident lymphatic infiltrations were seen in the lamina propria with scarce vascular swelling in the mucosa (**Fig 5**).

Studies on the 30th day revealed the progression of these lesions and necrosis to severity. The cryptic cell hyperplasia led to necrosis of the epithelial cells of the crypt with infiltrated lamina propria exhibiting necrotic variations like nuclear pyknosis and fragmentation. The graded score for the crypt epithelial necrosis was significantly higher (3.0 ± 0.00) than on the 15th day (1.7 ± 0.33) and also than that in controls (0.3 3± 0.33) whereas that of the necrotic lymphocytes in villous epithelium, Peyer's patches and lamina propria did not vary between groups.

*Group II.* Lesions were severe on both 15th and 30th days after toxin administration in this group. Similar vascular engorgement in the lamina propria, infiltration of the lymphoid cells

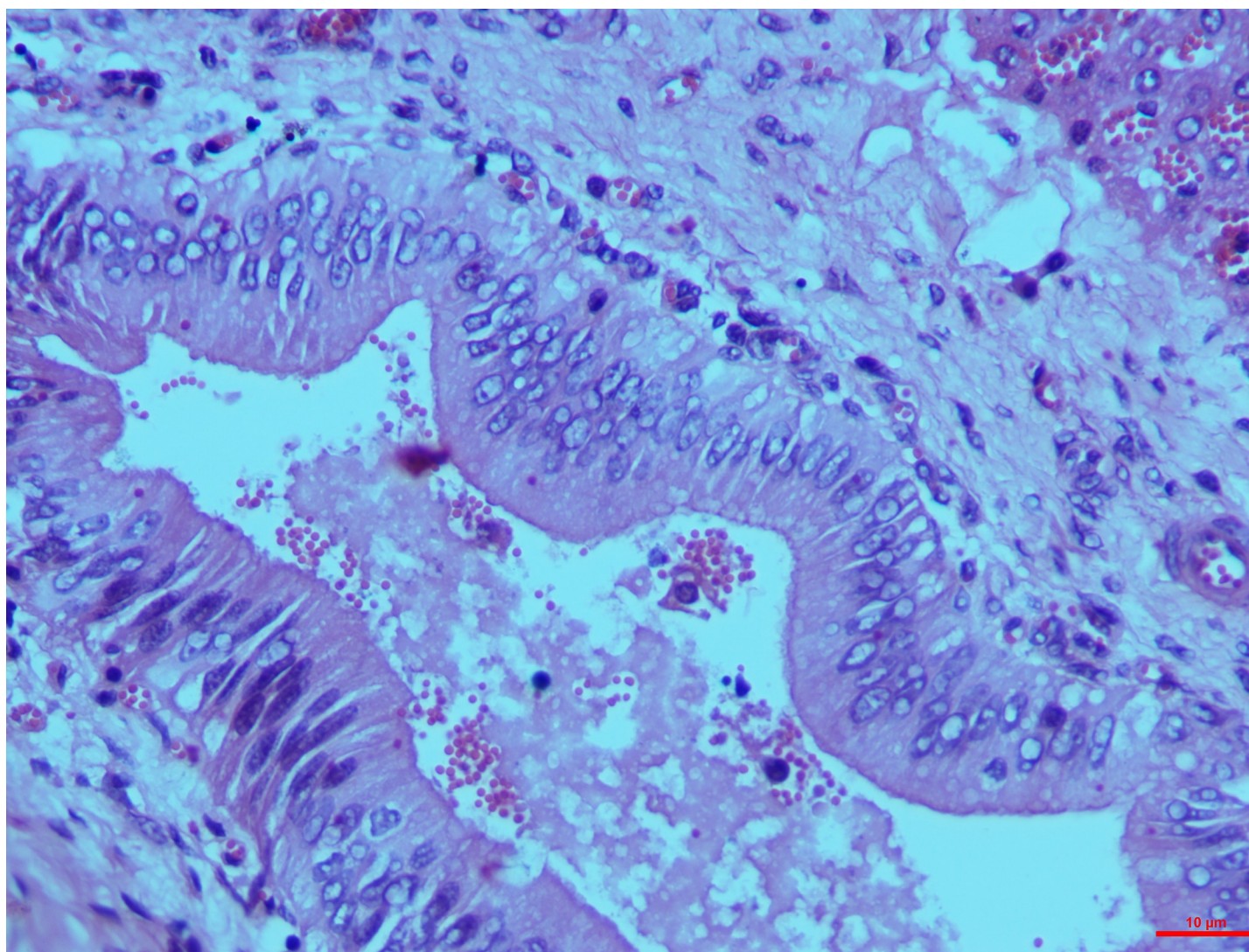

**Fig 1. Liver (group 1, 15 days): epithelial hyperplasia with bile ducts having degeneration of hepatocytes.** HE 400X.

like macrophages, lymphocytes and plasma cells in the inter-cryptic spaces and cell necrosis were observed on the 15th day (**Fig 6**).

They were progressive and became severe by the 30th day showing lymphocytolysis or depletion in the Peyer's patches along with necrotic fragmentation of nucleus and aggregation of the lymphoid cells in the lamina propria. In addition, the epithelial cells were diffused showing fragmented nuclei, infiltration and necrosis of the lymphocytes and disruption of the basement membrane were the visible features on the 30th day (**Fig 7**). The dead eosinophilic debris replaced the necrosed crypt epithelium with complete loss of cryptic architecture. The graded mean score of necrotic cells in lamina propria on the 30th day (2.7±0.33) was significantly higher when compared with that on the 15th day (1.33±0.66) and also than that in controls. But the score of necrosis in cryptic cells, villous epithelium and Peyer's patches did not differ significantly between days (Table 6).

**Mesenteric lymph nodes.** *Group I.* The histological changes were time-dependent, which means that the microscopic changes on the 15th day were less intense when compared to that

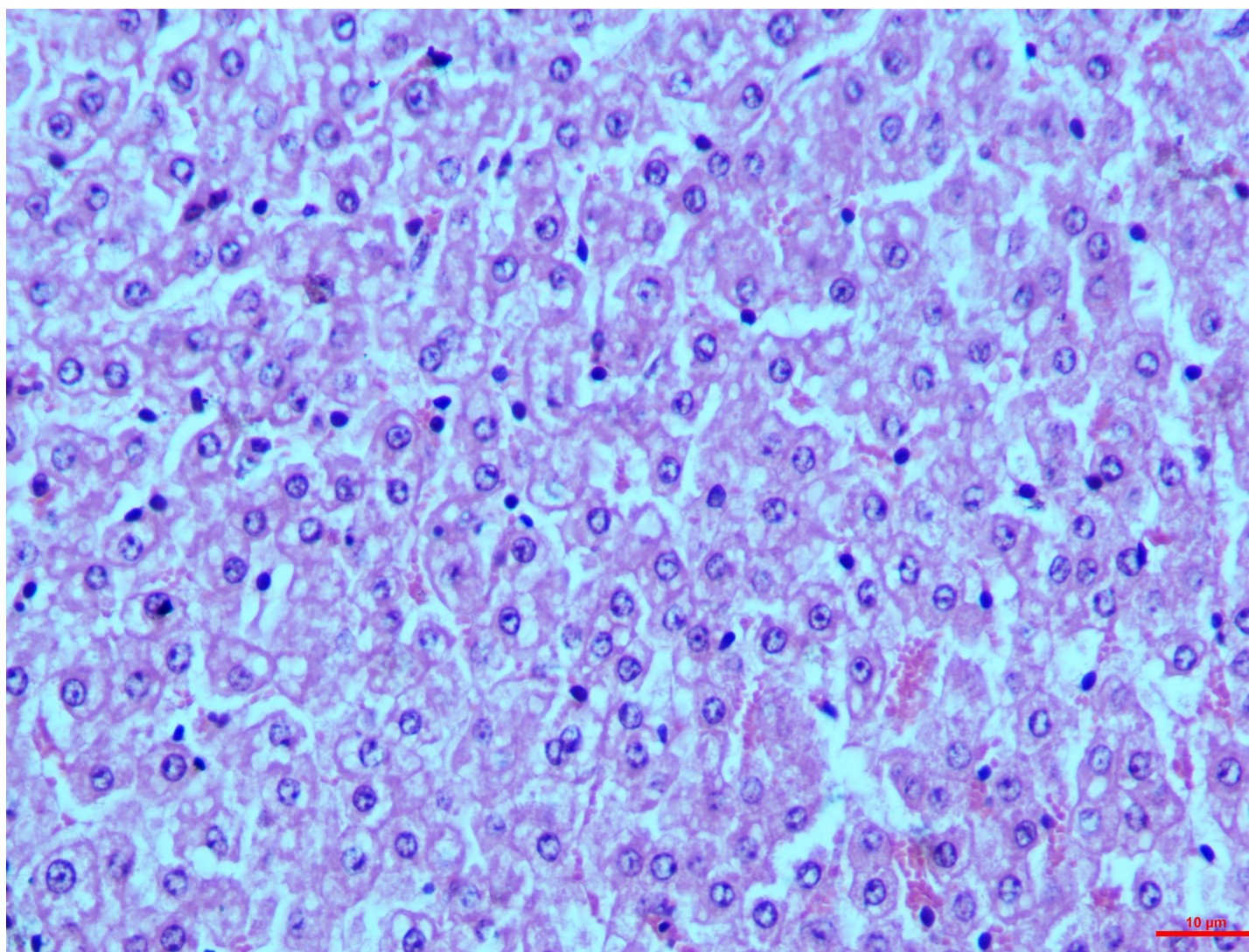

**Fig 2. Liver (group 1, 30 days): individual cell necrosis with dark pyknotic nucleus in the parenchyma with diffuses fatty changes of hepatocytes.** HE 100X.

of the 30th day. On the 15th day, mild depletion of the lymphoid follicles were seen with a scattered necrosis in the lymphoid cells whereas the changes on the 30th day included a widespread reduction in the cortical follicle and necrosis of the lymphocytes (Fig 8).

*Group II*. In the higher dosage treated group on 15th day, inter-follicular space exhibited moderate vascular swelling. In two of the animals, in the cortical zone infiltration of the polygonal cells was observed coupled with foamy cytoplasm similar to the epithelioid cells and macrophages. These changes progressed to severity on the 30th day showing severe depletion of the lymphoid cells and lymphocytolysis (Fig 9). These changes were characterized by scattered necrotic cells of the lymphocytes and macrophages in the lymphoid follicle of the cortex rendering it a starry sky appearance. All the three cases examined revealed signs of lymphoid depletion and infiltration in the polygonal cells to a large extent through the entire paracortical zone resembling epithelioid cells and macrophages. The graded score of necrotic cells was significantly higher in group II on 15th day (2.7±0.33) than in group I (1.00±0.00) and also than that in controls (0.7±0.33).

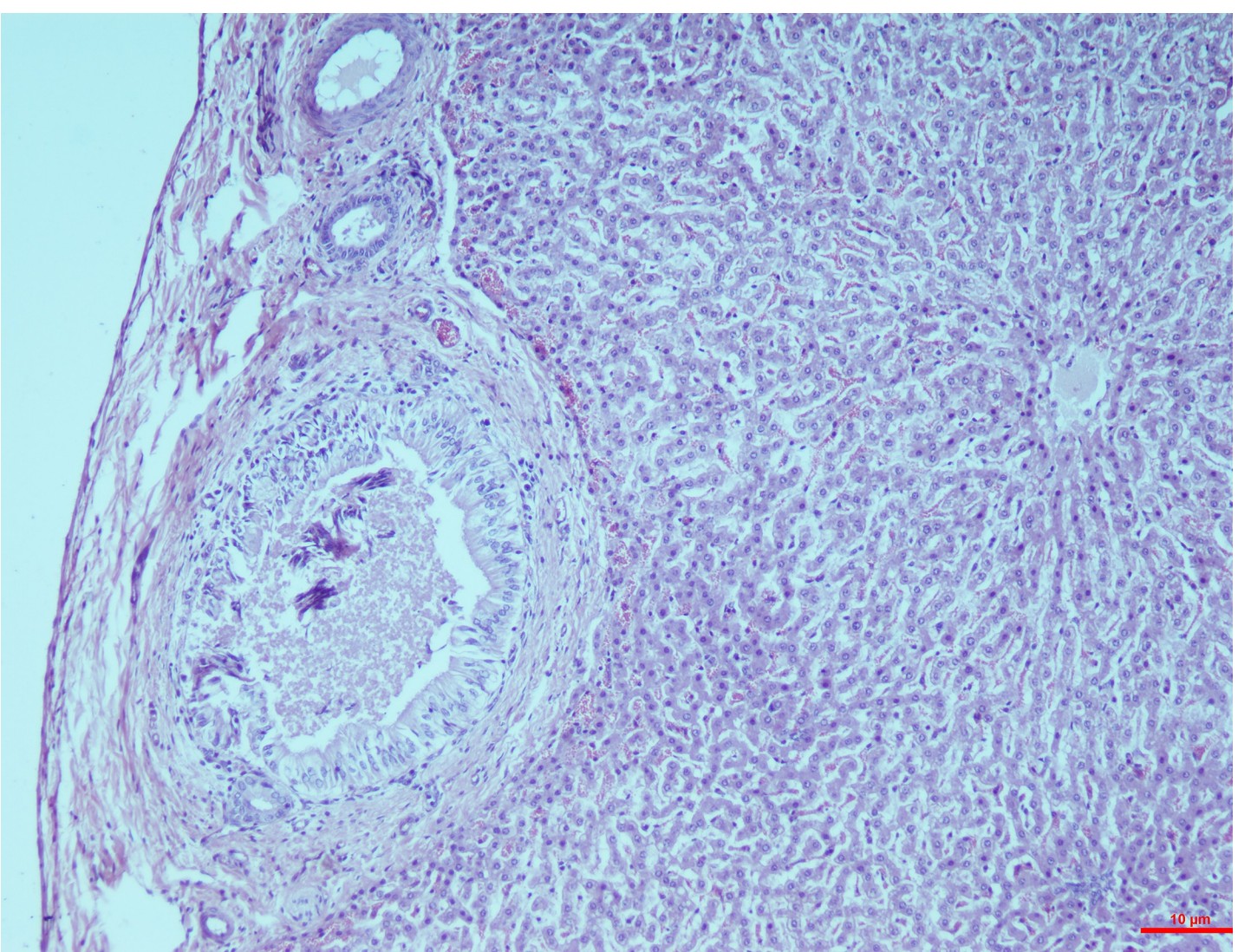

**Fig 3. Liver (group 2, 15 days): Hypertrophy peri-billiary hyperplasia/fibrosis with bile ducts having cell debris and diffuse centrilobular necrosis of hepatocytes.** HE 100X.

**Kidneys.** *Group I.* The histological changes in the kidney on 15th day were not remarkable but the pathology progressed over time. On the 30th day, features such as vascular engorgement, degeneration of the PCT and DCT with intact glomeruli were visible. Epithelial cells of the DCT showed detachment along with the degeneration of the vacuoles and occurrence of granular pink matter in the lumen. Similar to the other organs, the lesion progressed to necrosis in the kidney as well, with degeneration of the PCT and DCT along with the epithelial cell accumulation (Fig 10).

*Group II.* Characteristic epithelial cell damage showing eosinophilic granular cytoplasm was seen on the 15th day. Epithelial cells were swollen with haemorrhages in the interstitial cells in two of the cases. Likewise, complete epithelial destruction leading to its accumulation in the lumen was seen on the 30th which included loss of nuclei and brush border (Fig 11).

**Spleen.** *Group I.* The parenchyma cells of the spleen exhibited observable red pulp because of the engorgement of the sinusoid. Lymphocytes were slightly depleted in the marginal zone

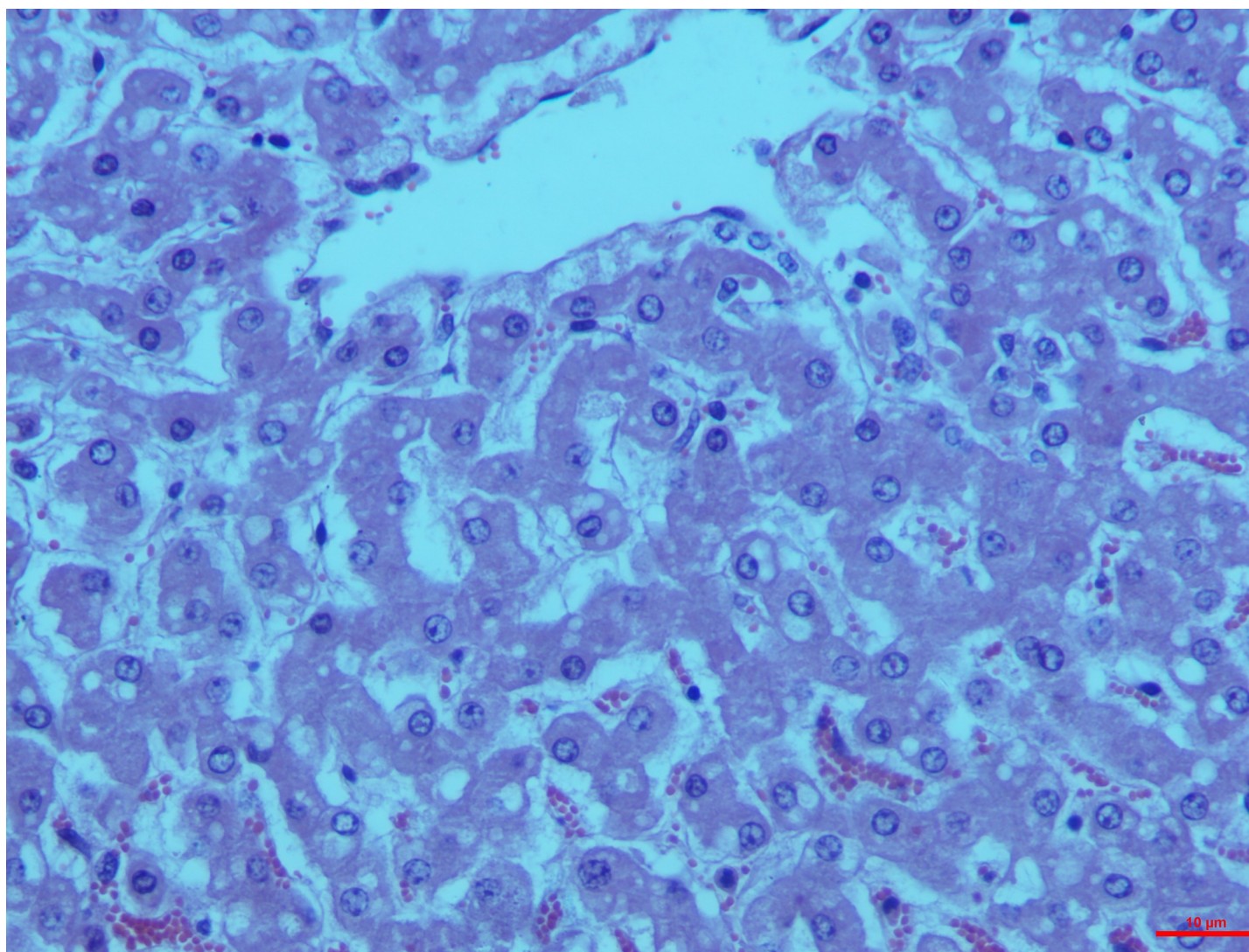

**Fig 4. Liver (group 2, 30 days): Hepatocytes showing distinct dark nucleus and homogeneous deep eosinophilic cytoplasm called as 'Ghost hepatocytes'.** HE 400X.

of the pulp. On the 30[th] day however, moderate depletion of the follicles and necrosis of the lymphocytes in the spleen was seen around the central artery (**Fig 12**). The graded mean score of lymphoid depletion (2.7±0.33) and necrosis (2.00±0.57) on 30[th] day was significantly higher than that in controls (0.33±0.33). Although there was little difference in the score between intervals in the toxin treated groups.

*Group II*. On the contrary to group I, group II showed severe depletion in the lymphoid cells right from the day 15 around the marginal and parafollicular zones. Two of the cases exhibited moderate necrosis also. Conversely, on the 30[th] day there was severe depletion in the lymphoid cells, thickening of the capsules and congestion in the sinusoid. It was also notably that prominent epithelioid cells reacted in the white pulp leading to lymphocytolysis. These cells were pleomorphic, large resembling macrophages in white pulp (**Fig 13**). Graded score of lymphoid depletion and necrosis of lymphoid cells was significantly higher on 15[th] and 30[th] day when compared to that in controls. There was no difference between intervals but higher score was recorded in toxicated groups than in the control group (Table 7).

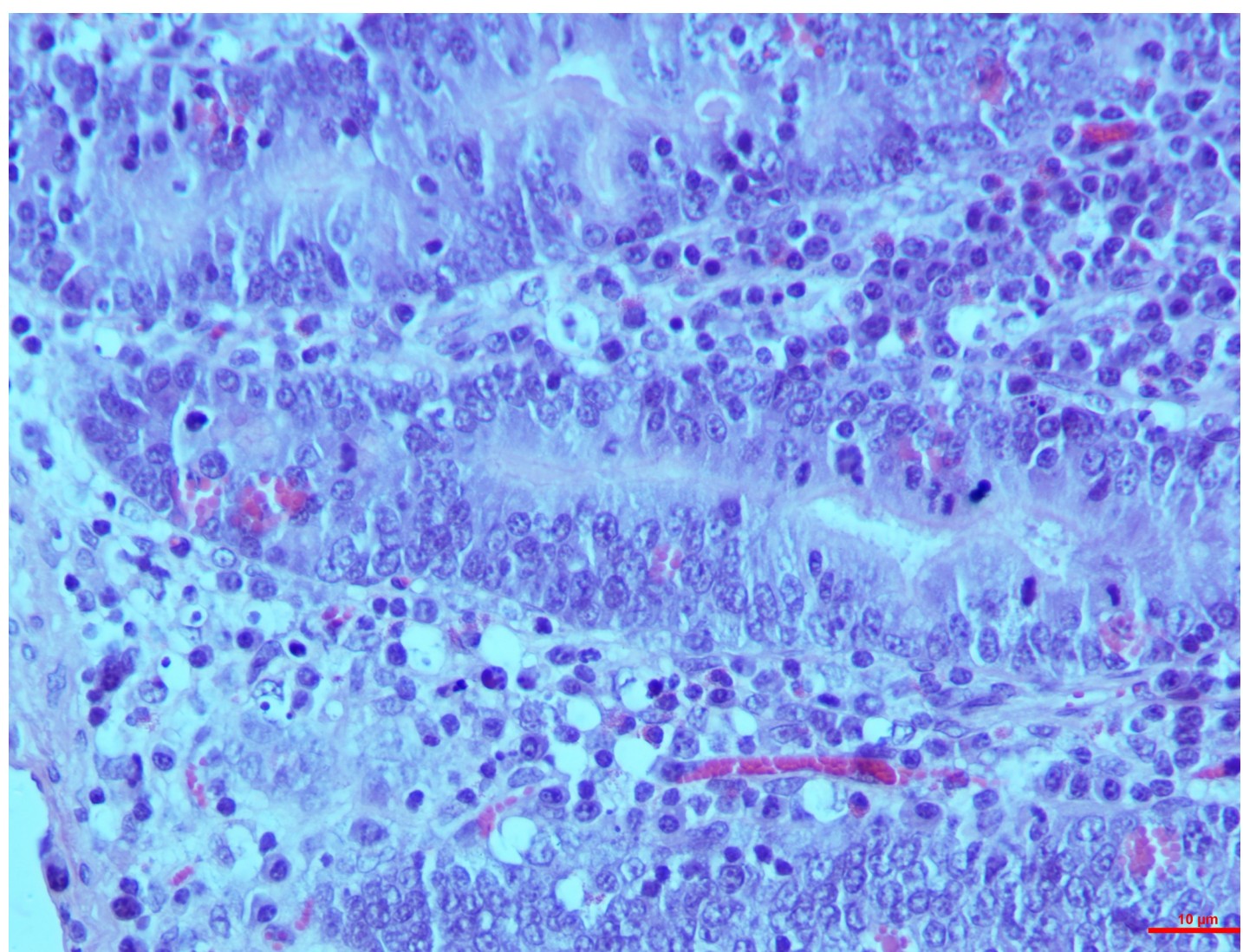

**Fig 5. Intestines (group 1, 15 days): Lamina propria showing diffusely infiltrated with necrotic lymphocytes, nuclear pyknosis and fragmentation.** HE 100X.

**Brain.** *Group I*. On 15th day, mild focal glial cell reaction was visible containing a few microglia and oligodendroglia with occasional neurons exhibiting shrinkage and degeneration. On 30th day, vascular changes such as micro capillaries engorgement and congestion in the cerebral cortex and thalamic region were observed in two cases. These cases also showed focal glial cell reaction with degeneration of cortical neurons having distinct dark nucleus.

*Group II*. On 30th day, neurons showed shrinkage and degeneration with eosinophilic cytoplasm. Dead neurons were surrounded by microglia and oligodendroglia cells indicating satellitosis and neuronophagia (**Fig 14**). Degeneration of Purkinje cells in cerebellum which was characterized by the loss of nucleus and homogenous amorphous cytoplasm was evident.

## Ultrastructural changes

**Liver.** Numerous vacuolation was observed close to the nuclei in the liver cells. These nuclei exhibited aggregation and margination of the heterochromatin with indistinct nuclear membrane. Mitochondria also showed loss of cristae and dissolution of the inner matrix and

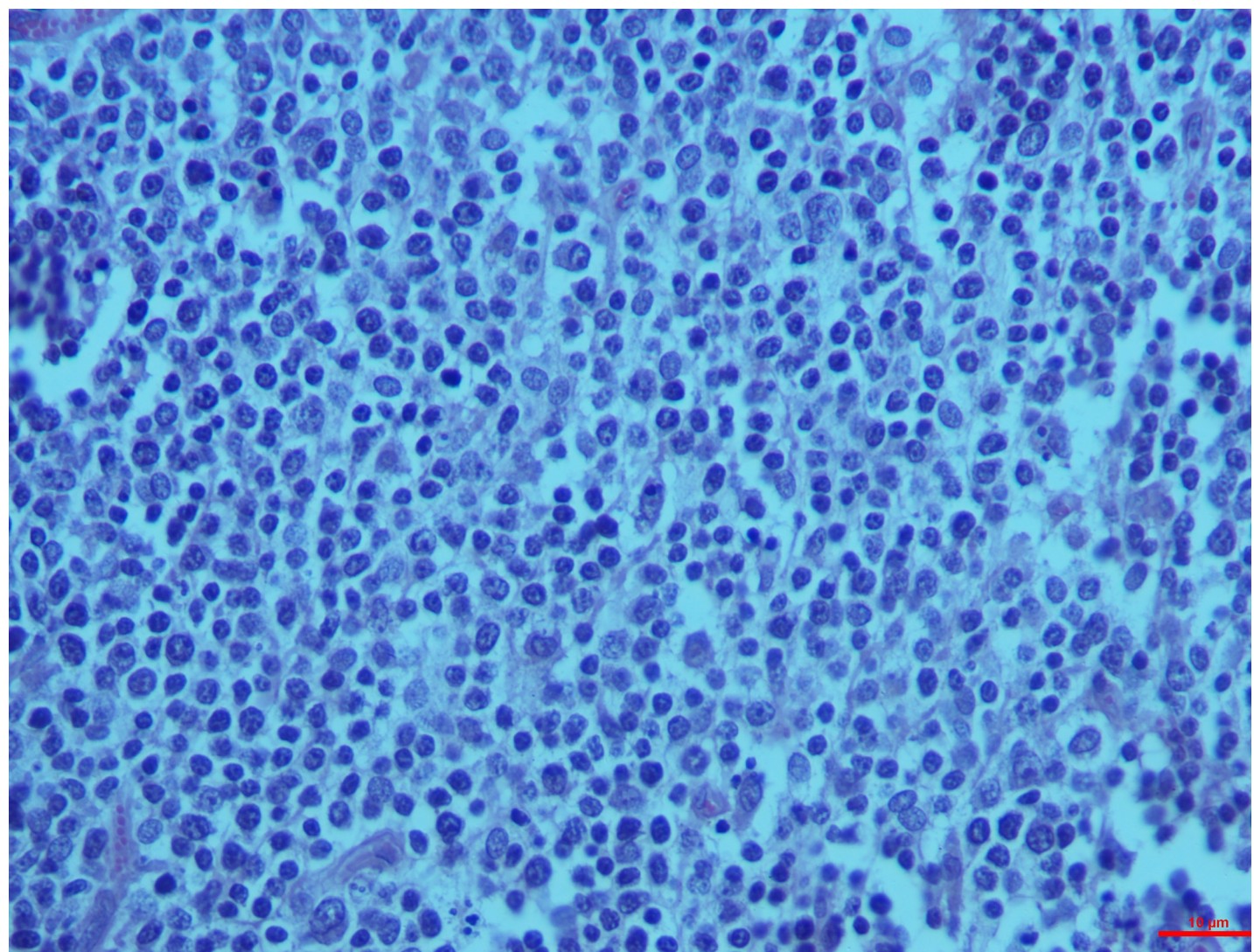

**Fig 6. Intestines (group 2, 15 days): Lamina propria showing depletion/lymphocytolysis in Peyer's patches.** HE 400X.

looked like an empty vacuole in group II after 30 days (**Fig 15**). While the cross sections in group II after 30 days showed severe hyperplasticity of the collagen (**Fig 16**).

**Small intestine.** Heterochromatin exhibited condensation, margination and clumping with indistinct nuclear membrane. Enterocytes showed consistent variations in both the groups on the 30[th] day. Degenerated and pleomorphic mitochondria with loss of cristae and dissolution of inner matrix was also observed (**Fig 17**).

**Kidneys.** Mitochondria and nucleus showed extensive degeneration on the 30[th] day. Epithelial cells were also affected showing loss of cristae leading to empty space and rendering the mitochondria pleomorphic. Condensation and margination of heterochromatin with indistinct nuclear membrane were also evident as shown in **Fig 18** in the higher dose group i.e. group II after 30 days ().

**Brain.** Neuronal degeneration was the most consistent finding noticed in group II on 30[th] day. It was characterized by presence of circular, whirling dark structures in the brain (**Fig 19**).

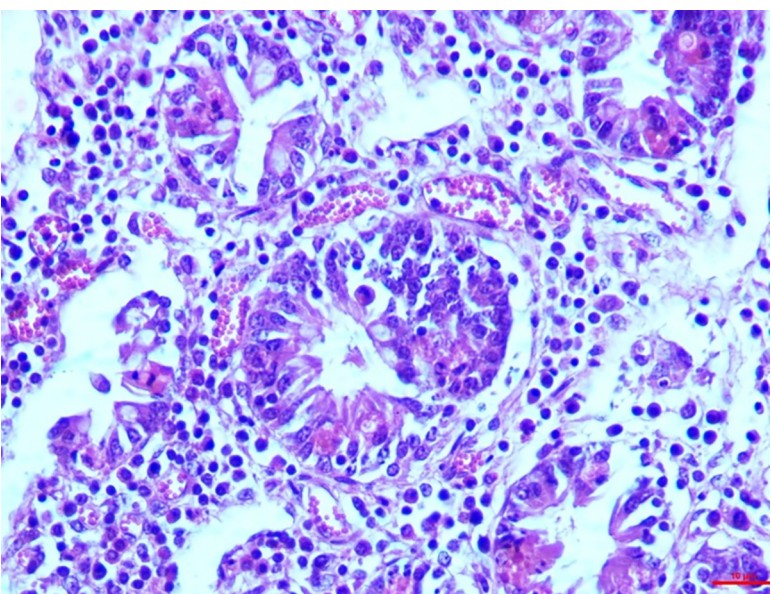

**Fig 7. Intestines (group 2, 30 days): Crypts in lamina propria showing diffuse epithelial necrosis with fragmentation of nuclei, disruption of BM and necrosis of lymphocytes.** HE 400X.

**Apoptotic changes.** Apoptosis was detected in hepatocytes around the central vein and portal triad in group I after 30 days (**Fig 20**). Within the intestine in group I after 30 days, apoptosis was detected in cryptic epithelial cells of the small intestine (**Fig 21**). Whereas in group II after 30 days, Peyer's patches of which the cryptic epithelial cells showed extensive apoptotic bodies (**Fig 22**) and within the MLN was also extensive apoptosis was witnessed in the lymphoid follicles (**Fig 23**). Within the kidney tissues, PCT and DCT epithelial cells exhibited apoptotic changes in group II after 30 days (**Fig 24**). The white pulp of the spleen also exhibited extensive apoptotic lymphoid cells in group II after 30 days (**Fig 25**). Among all the treated groups, group I and group II showed significantly higher score on the 30[th] day than on the 15[th] day when compared to controls. However, brain did not reveal the presence of any apoptotic cells in both the treatment as well as control groups. Mean score of TUNEL positive apoptotic cells in liver, intestine, MLN, Kidneys, spleen and brain on 15[th] and 30[th] day in both the groups were counted in high power field (x400) and presented in Table 8.

## Differential expression of Heat shock proteins, pro-apoptotic proteins and pro-inflammatory cytokines

Differential expression of mRNA of HSP-72, 90 and 27 genes were studied in liver, intestine, MLN, kidneys, spleen and brain in response to T-2 toxin in all the three groups. Overall, the

**Table 6. Graded score of necrotic cells in intestine in response to T-2 Toxin @ 10 ppm and 20 ppm on 15[th] and 30[th] day of toxicosis in goats.**

| Organ | Days | Crypt epithelium | | | Lamina propria | | | Villus epithelium | | | Peyer's patches | | |
|---|---|---|---|---|---|---|---|---|---|---|---|---|---|
| | | I | II | III | I | II | III | I | II | III | I | II | III |
| Intestine | 15[th] | 1.7±0.33[bB] | 3.0±0.00[a] | 0.33±0.33[c] | 1.0±0.57 | 1.33±0.66[B] | 0.7±0.33 | 1.0±0.57 | 0.7±0.33 | 0.33±0.33 | 1.0±0.57 | 1.33±0.33 | 0.33±0.33 |
| | 30[th] | 3.0±0.00[aA] | 2.7±0.33[a] | 1.0±0.57[b] | 1.7±0.33[b] | 2.7±0.33[aA] | 0.33±0.22[c] | 1.0±0.57 | 0.7±0.33 | 0.33±0.33 | 2.0±0.57[a] | 2.33±0.66[a] | 0.7±0.66[b] |

Group I-10 ppm, II-20 ppm and III- Control, Score: 1-mild necrotic cells, 2-moderate necrotic cells, 3- Sever necrotic cells. The data were analysed using one way and two way ANOVA. Mean bearing at least one common superscript did not differ between days and groups. Each part of intestine analyzed separately. The level of significance was kept at 0.05.

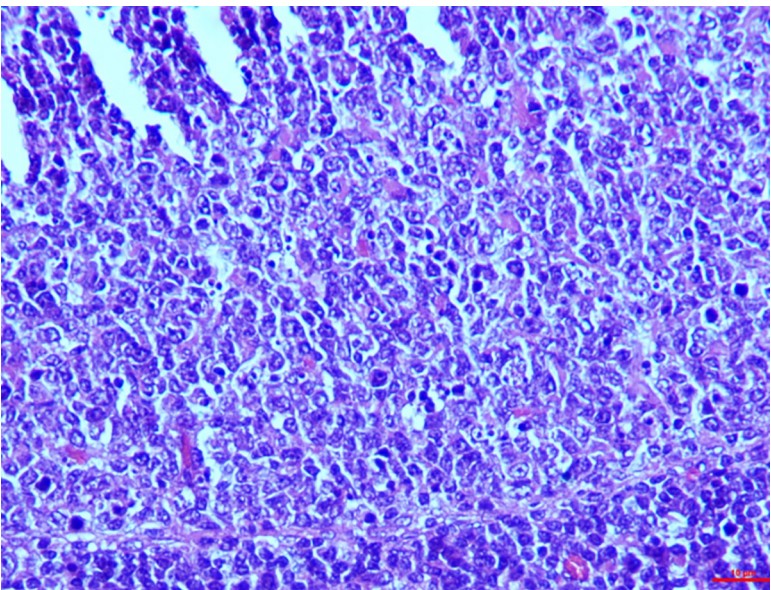

**Fig 8. Mesenteric lymph nodes (group 1, 30 days): Widespread depletion in cortical follicles with diffuse necrosis of lymphocytes/lymphocytolysis.** HE 400X.

mRNA expression of all three HSPs in the organs tested was significantly upregulated in group II when compared to group I, suggesting that the higher dose has led to increased expression of these genes as a result of stress exerted by the toxin (Table 9). Specifically, the expression of HSPs in the liver, intestine and MLN was higher in group II than in group I (Fig 1a). On the contrary, in the kidney expression of HSP-90 was significantly (p = 0.51) upregulated in group II compared to the group I, whereas HSP-72 and HSP-27 did not differ significantly (p = 0.51

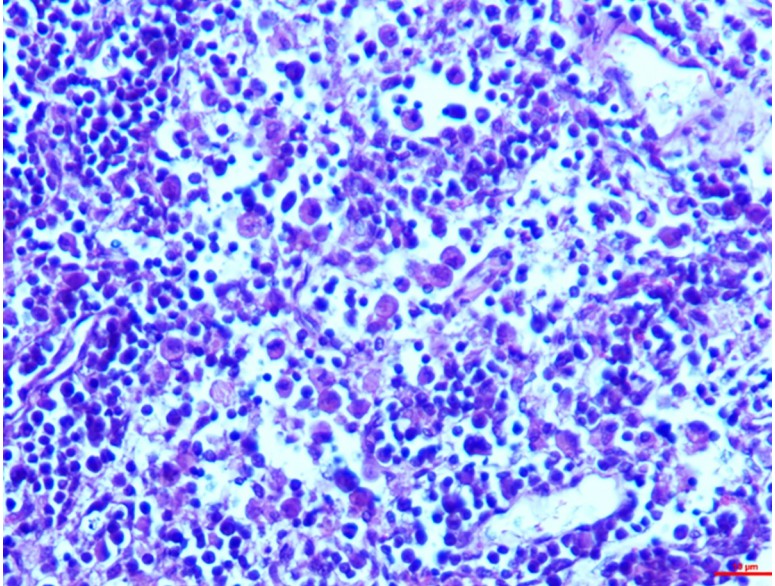

**Fig 9. Mesenteric lymph nodes (group 2, 30 days): Lymphoid depletion and infiltration of large polygonal epithelioid cells and macrophages.** HE 400X.

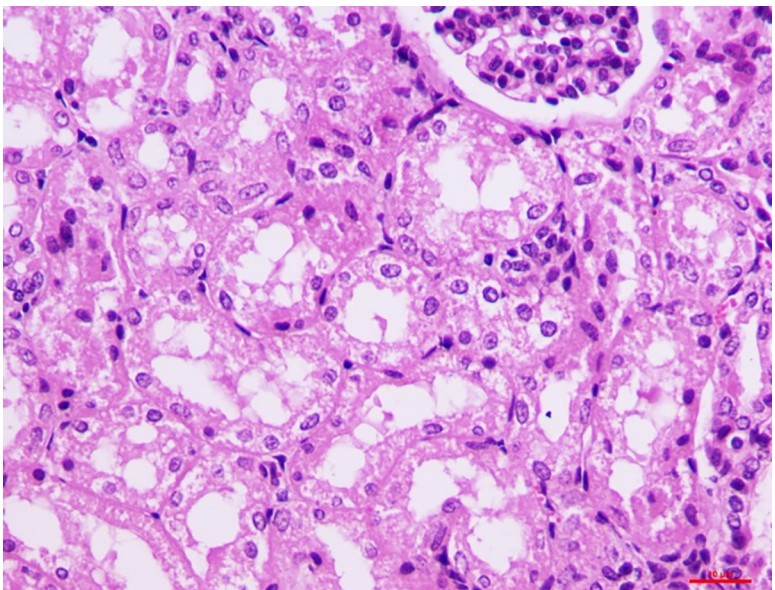

**Fig 10. Kidney (group 1, 30 days): Epithelial cells of DCT showing detachment, degeneration of vacuoles and occurrence of granular pink matter in the lumen.** HE 400X.

and -27 p = 0.27, respectively) between groups. Meanwhile, expression of HSP-72 and HSP-27 mRNA in the spleen was significantly high (p < 0.05) in group II than in group I. The expression of HSP-72 mRNA increased in group II than in group I on the 30th day. The expression of HSPs mRNA in brain did not differ between groups but showed up regulation than the control group.

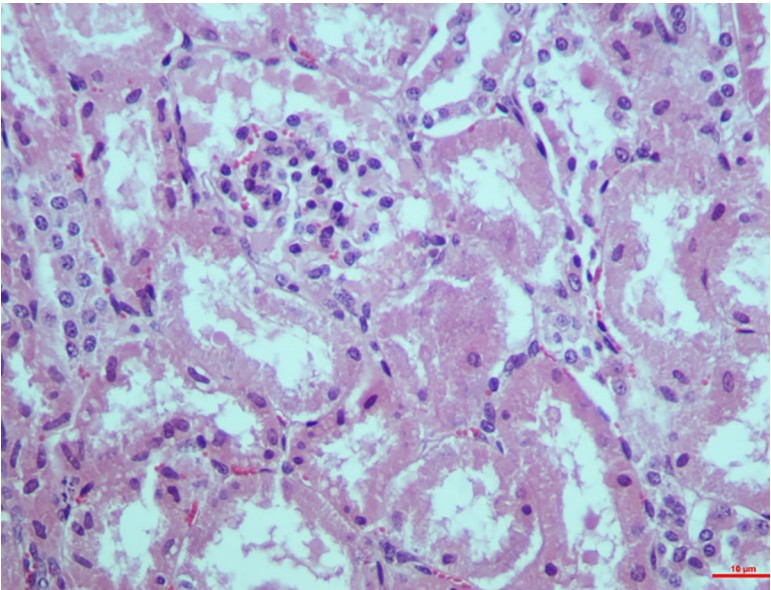

**Fig 11. Kidney (group 2, 30 days): Degeneration of PCT and DCT showing loss of nuclei and brush borders.** HE 400X.

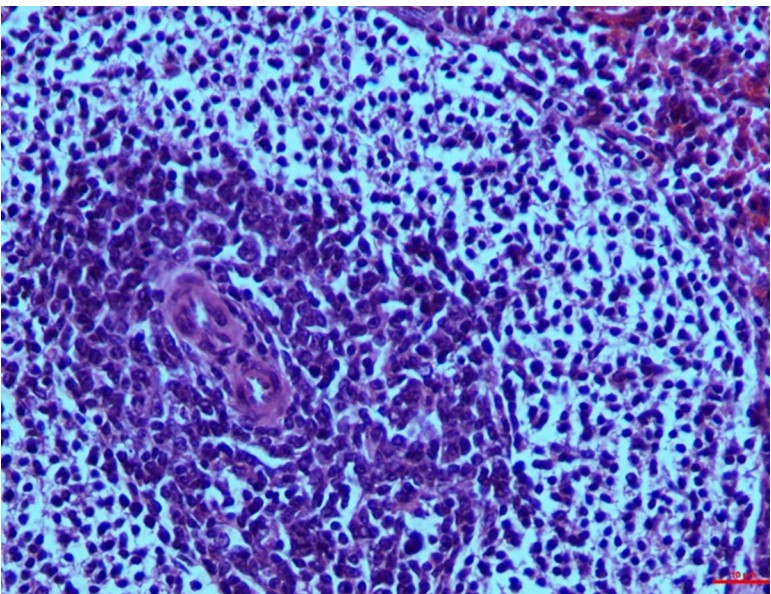

**Fig 12. Spleen (group 1, 30 days): Follicles showing depletion and necrosis of lymphocytes around the central artery.** HE 400X.

**Apoptotic genes.** Caspase-3 and Bax are pro-apoptotic proteins whereas Bcl-2 is an anti-apoptotic protein. Expression of these genes will clearly suggest the activation of apoptosis within the tissues due to the T-2 toxin induced toxicity. In our study, hepatic mRNA expression of Caspase-3, Bax was significantly ($p = 0.04$ and $p = 0.05$, respectively) higher in group I than in group II on 30th day, whereas that of Bcl-2 did not differ significantly ($p = 0.17$) between groups. However, in the intestine with caspase-3 and bax gene expression were

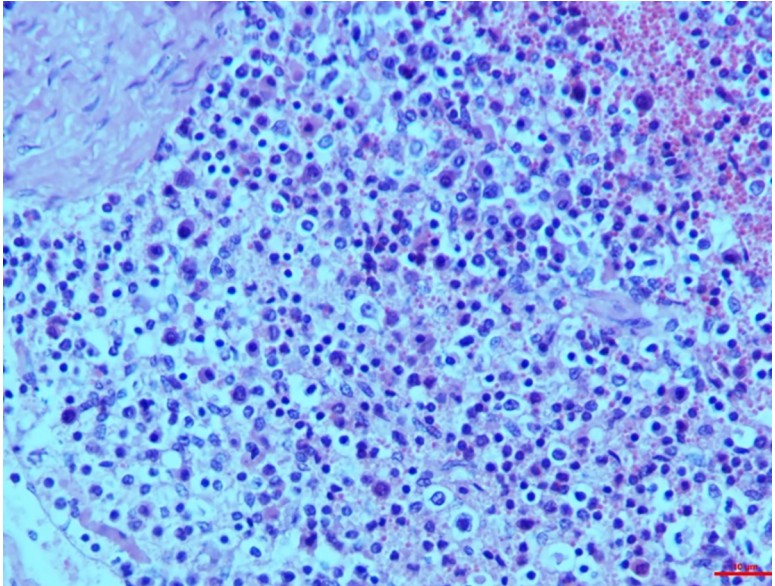

**Fig 13. Spleen (group 2, 30 days): Epithelioid cells reactions in the white pulp leading to lymphocytolysis.** HE 400X.

**Table 7. Graded score lymphoid depletion and necrosis of lymphoid cells (lymphocytolysis) in response to T-2 toxin @ 10 ppm and 20 ppm on 15th and 30th day.**

| Organ | Days | Lymphoid depletion | | | Lymphoid necrosis (lymphocytolysis) | | |
|---|---|---|---|---|---|---|---|
| | | I | II | III | I | II | III |
| Spleen | 15th | 1.33±0.33[b] | 3.0±0.00[a] | 0.7±0.33[c] | 1.0±0.57[b] | 2.7±0.33[a] | 0.7±0.33[b] |
| | 30th | 2.7±0.33[a] | 2.7±0.33[a] | 0.33±0.33[b] | 2.0±0.57[a] | 3.0±0.00[a] | 0.33±0.33[b] |
| MLN | 15th | 1.0±0.00[b] | 2.7±0.33[a] | 0.7±0.33[b] | 1.33±0.7[b] | 2.7±0.33[a] | 0.33±0.33[c] |
| | 30th | 2.0±0.57[a] | 3.0±0.00[a] | 0.7±0.33[b] | 2.33±0.7[a] | 3.0±0.00[a] | 0.33±0.33[b] |
| Peyer's patches | 15th | 1.0±0.00[a] | 1.7±0.33[a] | 0.33±0.33[b] | 0.7±0.33[bB] | 1.7±0.33[aB] | 0.33±0.33[b] |
| | 30th | 2.7±0.33[a] | 2.7±0.33[a] | 0.33±0.33[b] | 2.0±0.57[aA] | 3.0±0.00[aA] | 0.7±0.33[b] |

Group I-10ppm, II-20ppm and III- Control, Score: 1-mild depletion, 2-moderate depletion, 3- severe depletion. 1-mild necrotic cells, 2-moderate necrotic cells, 3-severe necrotic cells. The data were analysed using one way and two way ANOVA. Mean bearing at least one common superscript did not differ between days and groups. Each organ and parameter analyzed separately between groups and days. The level of significance was kept at 0.05.

significantly higher in group II than in group I (p = 0.001 and p = 0.05, respectively) and the expression of Bcl-2 was upregulated in group I than in group II (p = 0.05). On the contrary, the expression of caspase-3 was significantly higher in group II (p = 0.03) and the expression of Bax and Bcl-2 were significantly higher in group I than in group II within the MLNs (p = 0.57 and p = 0.04, respectively). Meanwhile, within the kidney, caspase-3 was significantly higher in group II (p = 0.04), whereas Bax was higher in group I (p = 0.03). Within the spleen, the expression of Bax and Bcl-2 was significantly higher in group I (p = 0.05) with no significant change in caspase-3 expression, whereas the expression of caspase-3 was significantly higher (p = 0.05) in group I within the brain. Collectively, it was observed that, T-2 toxin at most instances induced the expression of caspase-3 as well as Bax, with little or no significant changes in Bcl-2, indicating the induction of apoptosis within all the organs studied. (Table 10).

**Pro-inflammatory cytokines (IL-1α, IL-6 and TNF-α).** Various locally produced pro-inflammatory stimuli such as IL-1, IL-6 and TNF-α can invade into the circulation and induce

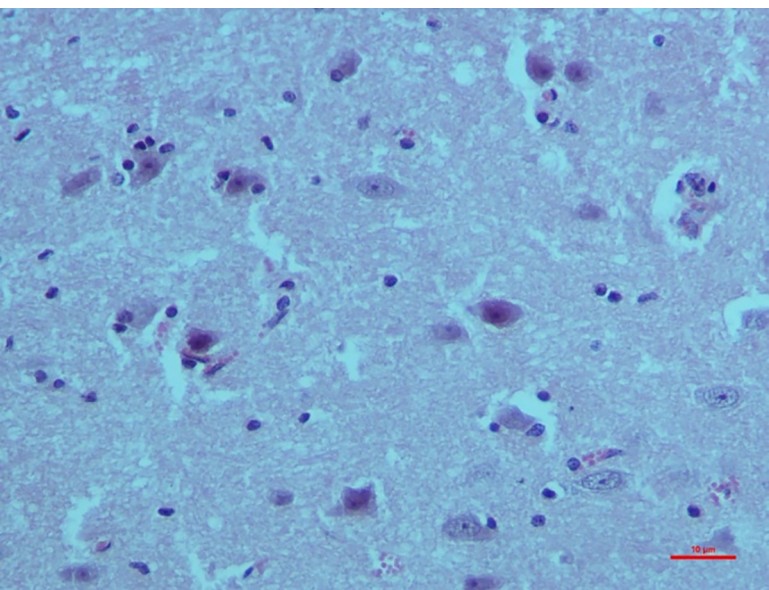

**Fig 14. Brain (group 2, 30 days): Neurons showing shrinkage and degeneration with eosinophilic cytoplasm indicating satellitosis.** HE 400X.

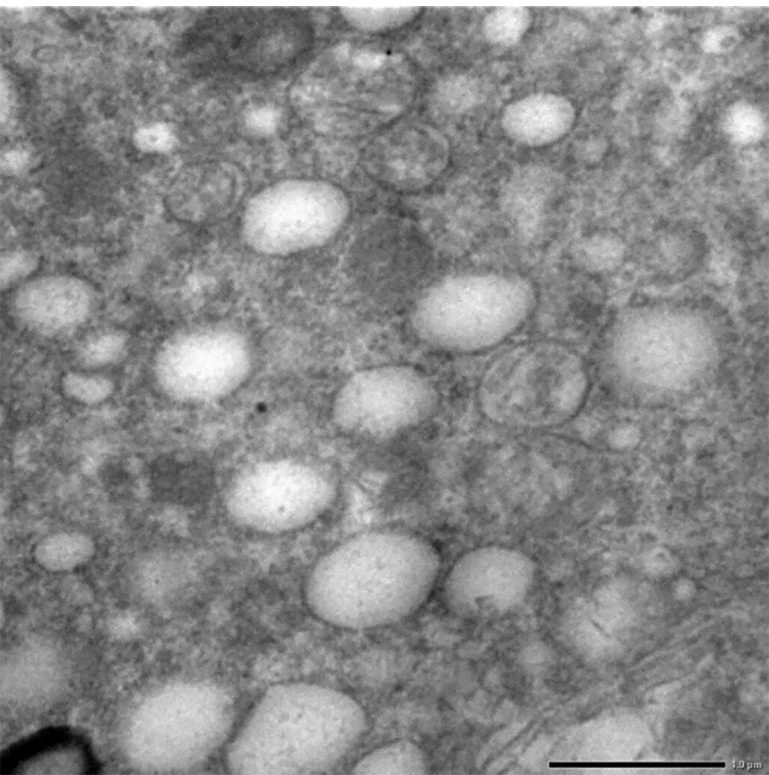

**Fig 15. Liver (group 2, 30 days): Mitochondrial degeneration with loss of cristae and formation of empty spaces in the cytoplasm of hepatocytes.** Uranyl acetate and lead citrate 200X.

a systemic response against the oxidative stress induced by the exposure to the toxin. Thus, in our study we assessed the levels of these mediators in various organs. It was observed that the hepatic expression of IL-1α, IL-6 (p = 0.04; p = 0.05, respectively) were significantly upregulated in group II, whereas the expression of TNF-α (p = 0.05) was upregulated in group I as against the that of the intestine where IL-1α expression was upregulated in group I but the expression of IL-6 and TNF-α were upregulated in group II. Similarly, only TNF- α (p = 0.04) significantly increased in group I within MLNs and IL-6 and TNF-α (p = 0.004 and p = 0.05) both varied in the kidneys. However, the expression of only IL-1α significantly (p = 0.05) increased in the group II animals within the spleen and brain whereas the expression of IL-6 and TNF-α did not exhibit much variation in the spleen and brain tissues. (Table 11).

## Discussion

In the present study, the experimental animals showed clinical symptoms such as diarrhea, feed refusal, lethargy and weakness after 15 days of toxin feeding at 10 and 20 ppm toxicity levels. The clinical signs exhibited by the animals in the present study were similar to those reported earlier by various workers in calves [38,39], sheep [18] and pigs[20]. Also, [18] reported that the lambs exposed to the T-2 toxin at a concentration of 0.6 mg/kg body weight for about 21 days exhibited periodic bouts of diarrhea and infection with Eimeria species, which are the clear symptoms of toxicosis. In addition, characteristics such as lethargy and feed refusal observed during the toxicosis might be due to the impairment in the central nervous system [40]. In some of the other studies, features of low feed consumption to complete feed refusal, reduced feed conversions for nutrient absorption, bloody diarrhea, reduction in

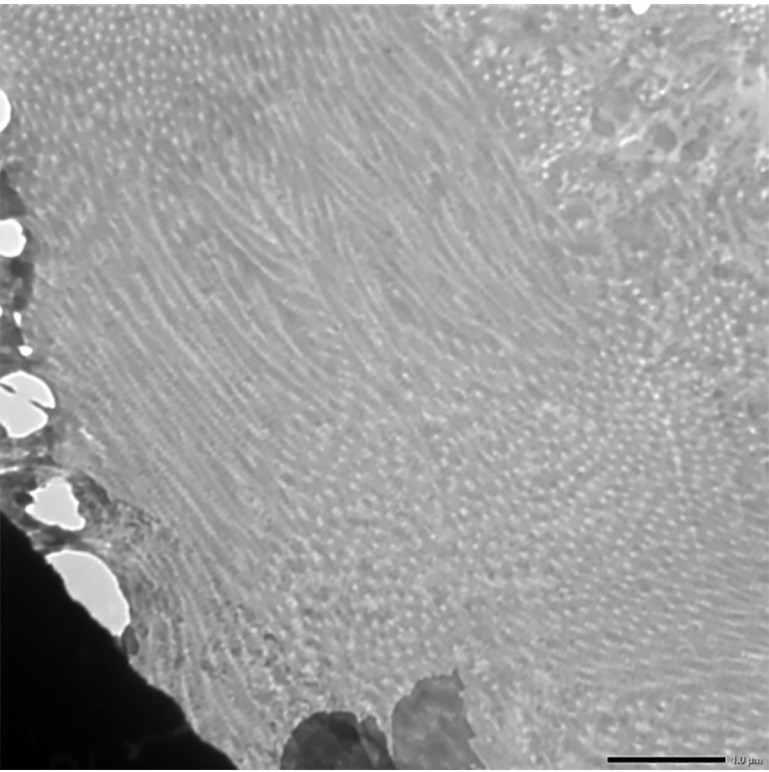

**Fig 16. Liver (group 2, 30 days): Hyperplasticity in the collagen fibers of liver.** Uranyl acetate and lead citrate
1000X.

milk production and absence of oestrus cycles were observed in the cattle with T-2 toxicosis
[16,38]. In addition, findings from the studies of [19] suggest that T-2 toxicosis in sheep (aged
1.5–3 yrs) showed similar clinical symptoms as stated above. The complete feed refusal
resulted in reduced body weight and in turn caused damage to the organs such as kidney, liver,
GI and lymphoid tissues [41]. Studies have also suggested that T-2 toxicosis-induced diarrhea
reduces the absorptive lining on the intestine and a corresponding reduction in the protein
synthesis [10,42]. All these changes might have led to a significant reduction in the body
weight of the experimental goats over a period of one month.

In our study, the dose- and duration-dependent reduction in the hemoglobin concentration
(greater at a high concentration) was observed, which is probably due to its role on hematopoi-
esis and inhibition of protein synthesis and is well studied in several animal models such as
rats, mice, guinea pigs, rabbits and cats [43,44]. The inhibition of hemoglobin synthesis can be
due to the reduced uptake of $^{59}$Fe by the erythrocytes which in turn lead to a reduced Hb pro-
duction [45,46]. In agreement with several other studies, severe leucopenia was also observed
in our study in the toxin treated groups around the 30[th] day [41,47], along with increased mye-
loid to erythroid ratios in lambs at a toxin concentration of 0.6 mg/kg body weight and 0.3
mg/kg body weight per day fed in a protein low diet for 21 days [18]. Along with its effects on
the hemoglobin content, the total platelet count was also reduced in our study on the 25[th] day
of toxin exposure, which might be a resultant of the direct effect on the platelet progenitor dif-
ferentiation and cytotoxicity [48]. Decreased platelet count was also reported in rabbits fed
0.5mg/kg T-2 toxin contaminated feed [47].

Group II on the 30[th] day exhibited significantly higher hepatic SOD and catalase enzymes,
suggesting the release of ROS, along with an increase in the hepatic LPO activity in group I in

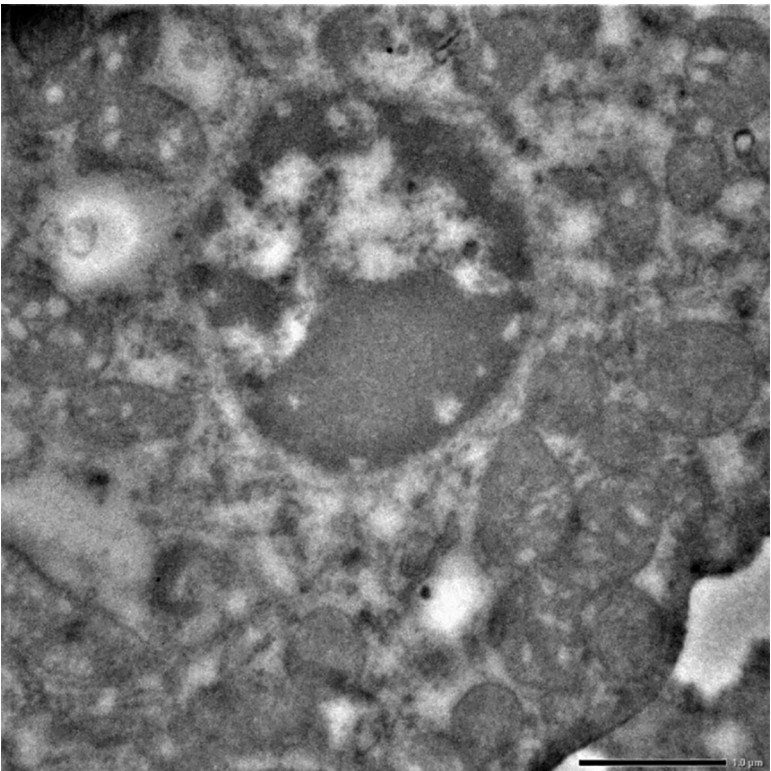

**Fig 17. Small intestine (group 2, 30 days): Heterochromatin showing condensation, margination and clumping with indistinct nuclear membrane in enterocytes.** Uranyl acetate and lead citrate 1000X.

the 15[th] day as well. [49]Reported that lipid peroxidation is not involved in T-2 toxin cytotoxicity. In contrast, [50] reported that T-2 toxin and deoxynivalenol (DON) stimulated lipid peroxidation in the liver of rats. These results suggest a direct effect of the toxin on the hepatic tissues at relatively lower doses administered for longer duration. It is a general understanding that oxidative stress is a resultant of either the generation of greater amount of ROS or reduced production of antioxidants within the cells [51,52]. Macromolecules critical to life namely the nucleic acids, lipids and proteins are the major targets of ROS, which further lead to tissue injury [53]. Oxidative damage around the 30[th] day post toxin exposure also led to severe histological alterations in the intestine including tissue necrosis of the epithelial lining and the lymphocytes in the lamina propria.

Apart from the generation of ROS, lipid peroxidation is also targeted by the toxin. However, reports on the T-2 toxin induced lipid peroxidation in animals are contradicting. [49] reported that lipid peroxidation is not involved in T-2 toxin cytotoxicity, whereas [50] reported that T-2 toxin and deoxynivalenol (DON) stimulated lipid peroxidation in the liver of rats. Some of the other studies report an increase in the lipid peroxide levels in the organs such as kidney, liver, thymus, bone marrow and spleen upon a single oral dose treatment of T-2 toxin @ 2 or 3.6 mg/kg body weight [48,53]. Similar to our findings, [33] reported an increase in the melonaldehyde, SOD and catalase enzymes in the liver and kidney tissue homogenates of the experimental rabbits after oral feeding of AFB1 @ 0.5ppm, ochratoxin @ 1ppm and in combination in rabbits on 30[th] and 60[th] day. Lipid peroxidation caused by T-2 toxin in liver and kidneys has been identified as an important underlying mechanism of T-2 toxin-induced cell injury, DNA damage and apoptosis [54]. On the contrary, the studies carried out by [55] demonstrated a

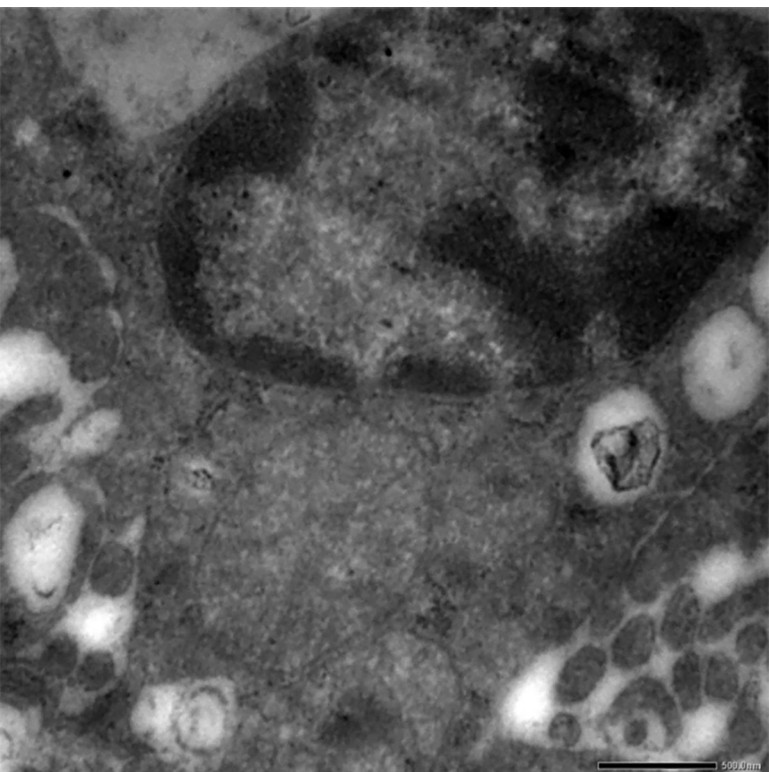

**Fig 18. Kidney (group 2, 30 days): Epithelial cells showing loss of cristae leading to mitochondria pleomorphic.**
Uranyl acetate and lead citrate 500X.

marked reduction of SOD and catalase activity along with a concomitant increase in the LPO levels in the liver and kidney tissues after oral treatment of T-2 toxin @ 0.5ppm, 0.75ppm and 1ppm in rats.

Considering the fact that any foreign particle from the feeds should enter the body through the GI system, it was evaluated in this study. The lesions were observed on 15 and 30[th] day showing mild crypt epithelial hyperplasia, infiltration of lymphocytes and vascular engorgement in the mucosa which progressed to severe necrosis of crypt epithelia, lymphocytes in lamina propria, infiltration of epitheliod cells, macrophages and plasma cells in group II on 30[th] day. Similar lesions in gastrointestinal system have been described in various livestock species by several workers in trichothecenes toxicity [11–15,38]. Such lesions also caused damage to the actively dividing cells lining the intestinal mucosa and leukocytes, suggesting the effects of the toxin on the intestinal cells [8]. Yet another study examined the T-2 toxicosis in piglets which showed congestion, hemorrhages and necrosis of the mucosal cells of GI and a direct effect on the lymphocytes and lymph nodes of the immune system. In a similar study on the pigs, [56] showed an increased endothelial lymphocytes and plasma cells of the epithelium along with a reduced number of goblets and underdeveloped glycocalyx in the epithelium of the intestine.

In the present study, the graded mean score of necrotic cells in crypt epithelium was significantly higher in group II but villous epithelium and LP did not show severe necrosis with low mean score of necrotic cells, which suggested that the cryptic epithelial cells were the primary targets for T-2 toxin exposure in intestine as this region consisted actively dividing cells [20]. Examination of the Peyer's patched showed severely diffused lymphocytes and depletion of the lymphoid in the higher dosage group, which is in agreement with the histological findings from the study on piglets treated with T-2 toxin @ 1.5mg, 2mg and 2.5mg/kg of BW [8].

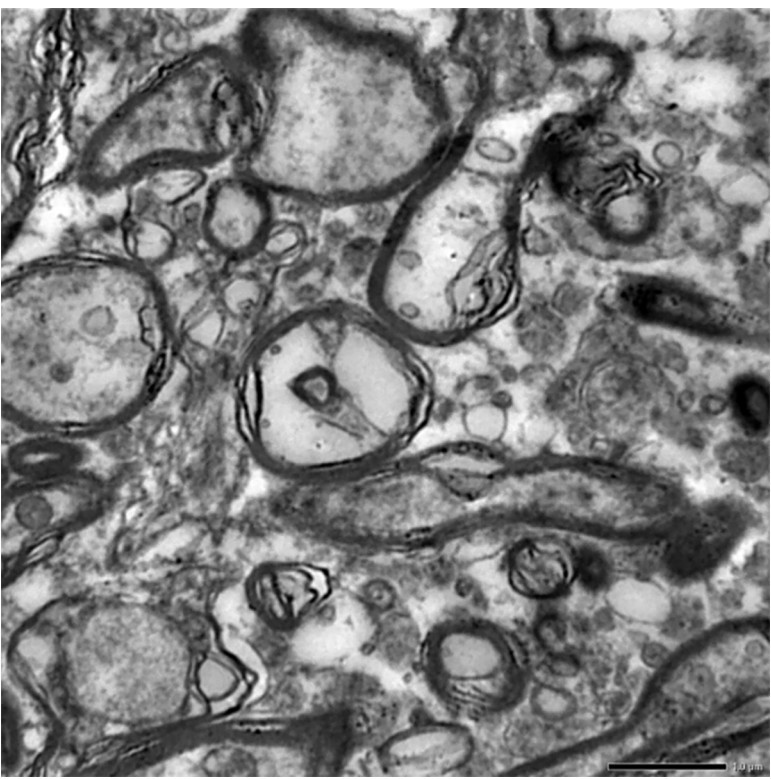

**Fig 19. Brain (group 2, 30 days): Degeneration of neurons by formation of circular, whirling dark structures called as myelin figures.** Uranyl acetate and lead citrate 1000X.

In addition, the epithelium of the intestine also revealed severe apoptotic modifications with a marked increase in the TUNEL positive cells in group II on the 30th day, which are also in agreement with the studies of Quiroga et al. [20] that described similar apoptotic changes in

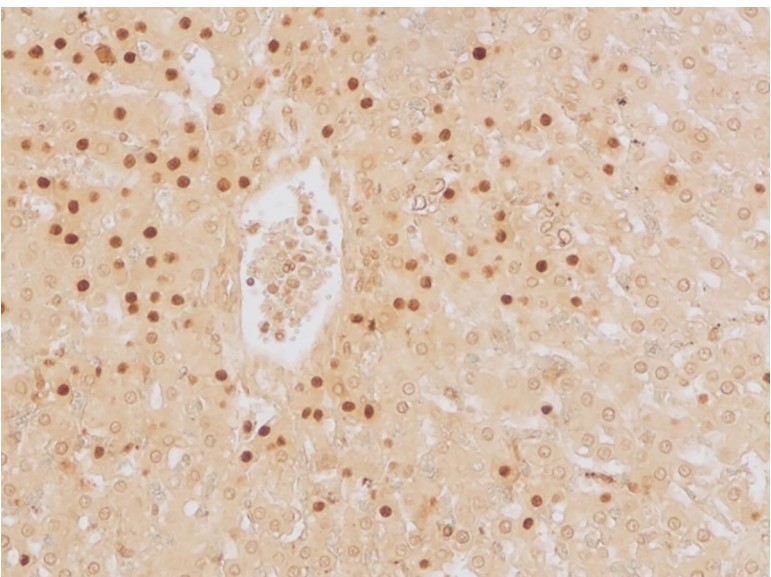

**Fig 20. Liver (group 1, 30 days): Apoptosis in hepatocytes around central vein and portal triad.** DAB 400X.

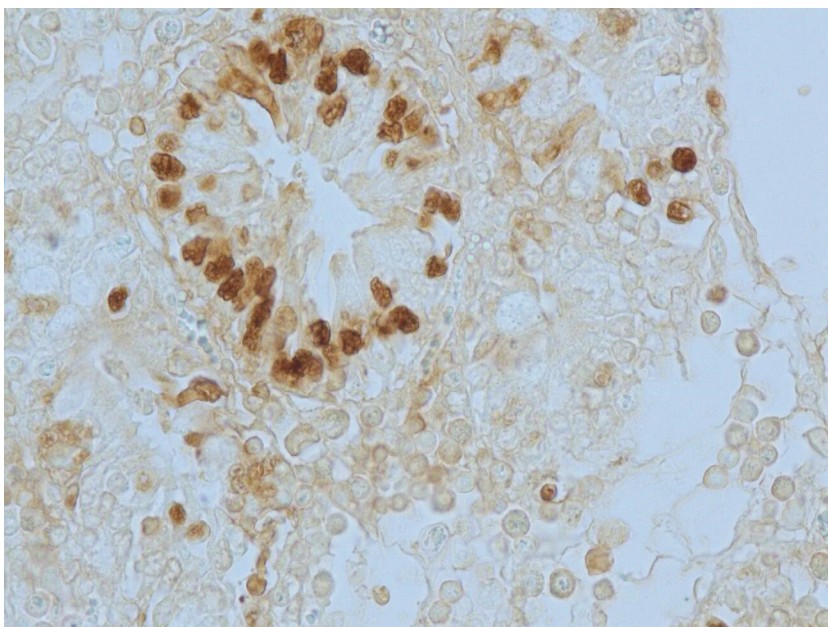

**Fig 21. Small intestine (group 1, 30 days): Apoptosis in cryptic epithelial cells of the small intestine.** DAB 400X.

the intestinal crypt cells. It can thus be inferred from grading the apoptotic changes in the Peyer's patches, intestinal crypts and lymphocytes that the intestinal lesions are in goats have similar pathogenesis as that in pigs [20].

Significant pathomorphological changes were also observed in the lymphoid organs (spleen, MLN and Peyer's patches) in our present study. Our study demonstrated remarkable enlargement of the MLN post toxin treatment along with lymphoid necrosis and depletion in a dose-dependent manner. Similar findings were shown by [18] during T-2 toxicosis in sheep.

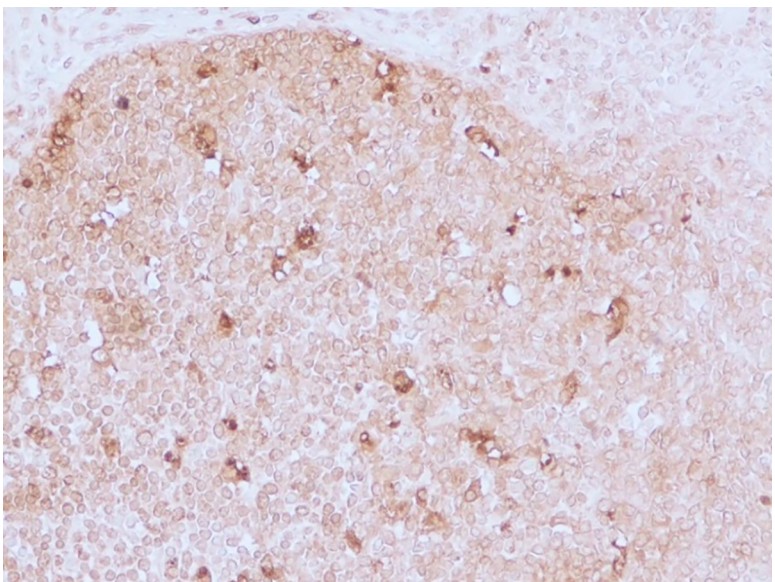

**Fig 22. Small intestine (group 2, 30 days): Peyer's patches in intestine showing apoptotic lymphocytes.** DAB 400X.

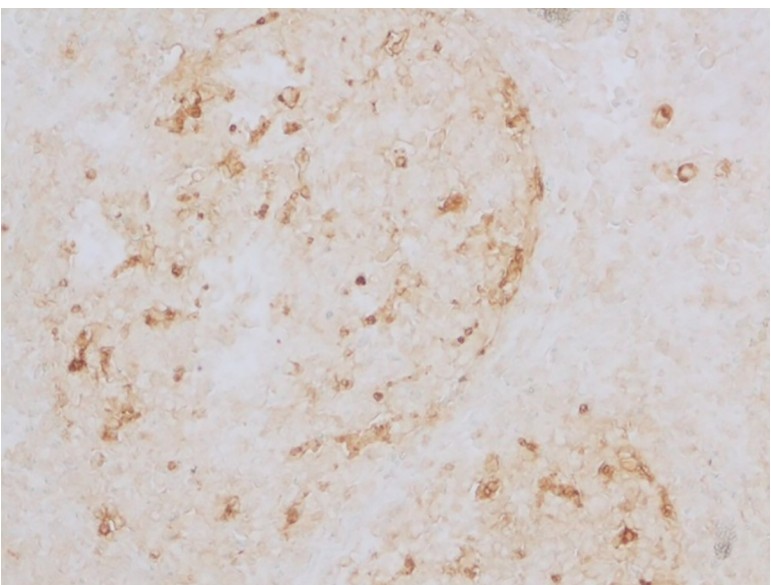

**Fig 23. Mesenteric lymph nodes (group 2, 30 days): Apoptotic lymphocytes in the follicles of MLN.** DAB 400X.

They study demonstrated depletion of lymphocytes in the medullary and paracortical areas and germinal centers, presence of numerous pyknotic cells in the MLN and hypocellularity in the germinal centers of spleen and were similar to T-2 toxicosis in sheep [18], piglets [20] and AFB1and ochratoxin toxicity in rabbits [33]. In support to these findings, [43] demonstrated higher concentrations of the toxin in the lymphoid organs as early as 3 h after the intra-aortal administration in the experimental pigs. Such a fast penetration could be attributed to its lipophilic nature facilitating it to surpass the blood-brain barrier [9].

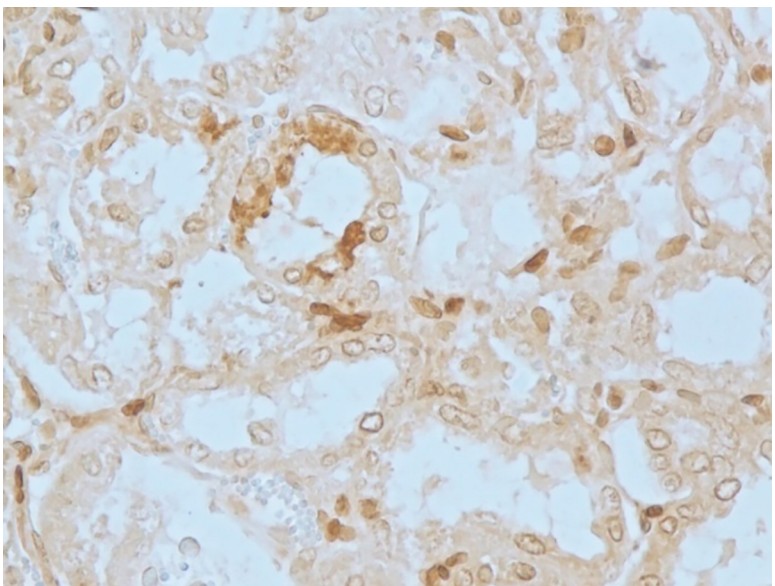

**Fig 24. Kidney (group 2, 30 days): PCT and DCT showing apoptotic epithelial cells.** DAB 400X.

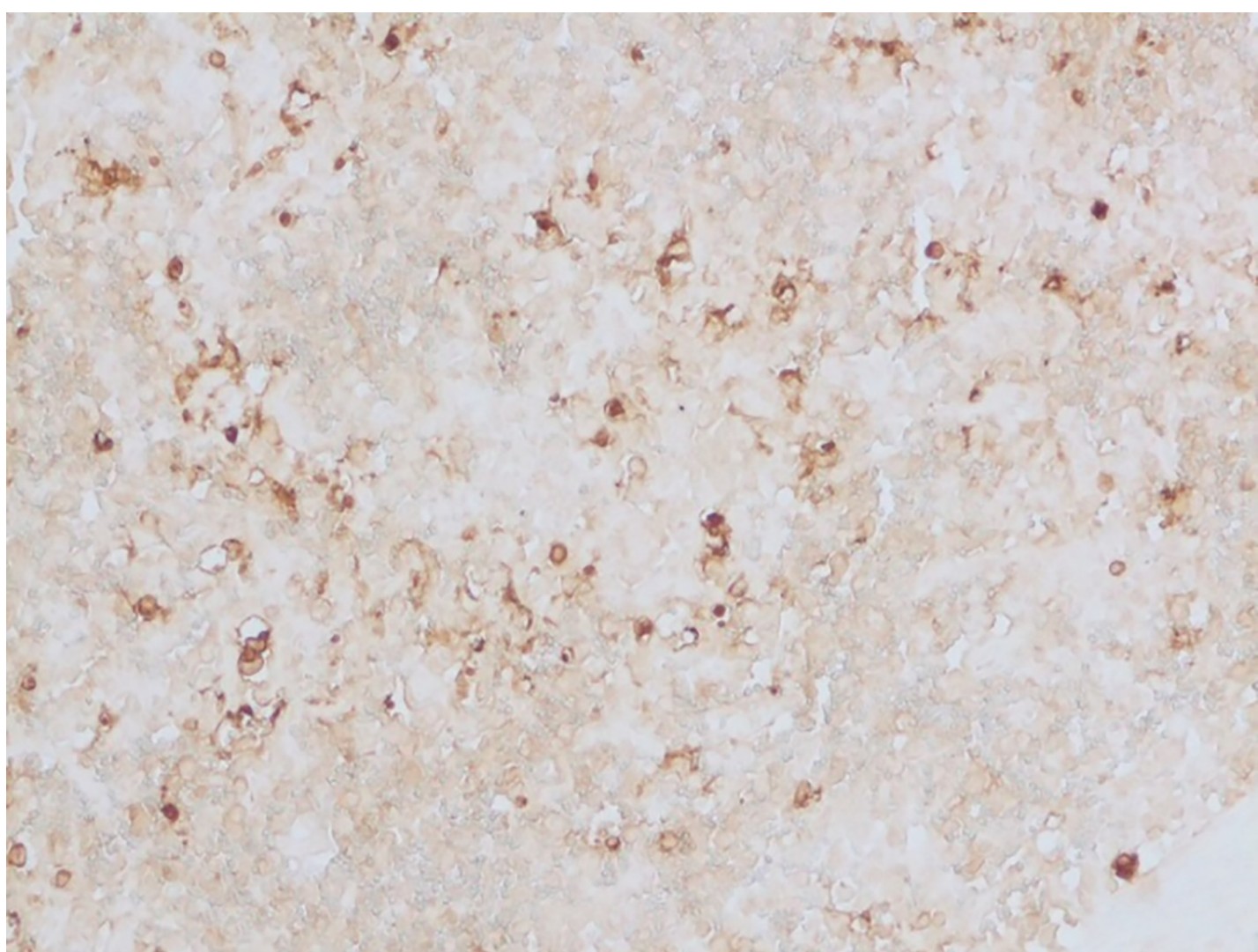

**Fig 25. Spleen (group 2, 30 days): Apoptotic lymphocytes in white pulp area of spleen.** DAB 400X.

In addition to the lymphoid organs, kidneys have shown marked pathomorphological changes including degeneration of the epithelial lining of proximal and distal convoluted tubules, interstitial engorgement, renal tubular necrosis and accumulation of the pink stained material in the lumen. The dose- and duration-dependent modifications in the kidney tissues indicate the effect of toxin on these tissues. The harmful effects of the toxin is either via the generation of oxidative stress, as observed by the increased stress enzymes, or because of the renal damage rendered during the biotransformation of the toxin and its excretion as observed in the studies conducted by Konigs et al [57] and Wu et al [58].

Besides these organs, T-2 toxicosis has also affected the brain tissues demonstrated by a varying degree of characteristic lesions including engorgement of the vascular tissues, Purkinje and cortical cell neuronal degenerations, focal glial reaction and satellitosis in both the toxin treated groups on 30th day. Along with these characteristics, enhanced lipid peroxidation and greater levels of oxidative stress enzymes in the brain tissues. However, TUNEL assay could not detect apoptosis in the brain cells. Effects of T-2 toxicosis on the neuronal cells in the brain have not been well studied. The neurological effects of the T-2 toxins are thought to be a result

**Table 8. Mean score (mean±SEM) of TUNEL positive (apoptotic) cells in different tissues in T-2 toxin induced apoptosis in toxicated and control groups on 15th and 30th days.**

| Organs/tissues | (Days) | Groups | | | F value |
|---|---|---|---|---|---|
| | | 10ppm (Group I) | 20ppm (Group II) | Control (III) | |
| Liver | 15 | 10.58±3.3$^{aB}$ | 5.33±1.6$^{aB}$ | 1.64±0.52$^{b}$ | $F_G$ = 36.78** |
| | | | | | $F_D$ = 13.78** |
| | 30 | 27.7±3.7$^{aA}$ | 21.7±2.5$^{aA}$ | 1.2±0.42$^{b}$ | $F_{DG}$ = 3.68** |
| Intestine | 15 | 4.7±1.17$^{bB}$ | 16.6±2.6$^{aB}$ | 1.8±0.32$^{b}$ | $F_G$ = 73.69** |
| | | | | | $F_D$ = 50.56** |
| | 30 | 20.7±2.2$^{bA}$ | 40.7±4.2$^{aA}$ | 1.2±0.30$^{c}$ | $F_{DG}$ = 14.18** |
| MLN | 15 | 4.8±0.9$^{abB}$ | 6.1±1.1$^{aB}$ | 1.8±0.6$^{b}$ | $F_G$ = 39.64** |
| | | | | | $F_D$ = 41.13** |
| | 30 | 10.5±2.1$^{bA}$ | 22.2±1.9$^{aA}$ | 2.1±0.9$^{c}$ | $F_{DG}$ = 17.3** |
| Kidneys | 15 | 3.7±1.2$^{bB}$ | 9.2±1.3$^{aB}$ | 0.9±0.3$^{b}$ | $F_G$ = 33.8** |
| | | | | | $F_D$ = 29.2** |
| | 30 | 14.7±1.4$^{bA}$ | 21.7±3.6$^{bA}$ | 1.22±0.43$^{c}$ | $F_{DG}$ = 7.38** |
| Spleen | 15 | 4.6±0.19$^{abB}$ | 7.1±1.2$^{aB}$ | 3.1±0.7$^{b}$ | $F_G$ = 14.3** |
| | | | | | $F_D$ = 13.08** |
| | 30 | 12.5±1.3$^{aA}$ | 9.3±1.4$^{aA}$ | 2.9±0.9$^{b}$ | $F_{DG}$ = 6.39** |

The data were analyzed using one way-ANOVA, two way-ANOVA and independent (t) test, wherever applicable. Superscripts A, B shows differences in columns and a, b, c in rows. Mean bearing at least one common superscript didn't show any difference. $F_D$ = effect within days (D), $F_G$ = effect within groups (G), $F_{DG}$ = interaction within days and groups *<0.05, **<0.01.

of either the metabolic changes in the biogenic mono amines [59] or changes in the permeability of amino acids across the blood-brain barriers [60].

Further, at the cellular level the modifications varied with respect to the organs, namely, liver, intestine, kidneys and brain. The nuclear membrane remained indistinct along with loss of cristae and inner matrix leading to valuolation of the mitochondria, hyperplasia and dilation of the rough endoplasmic reticulum and condensation of the heterochromatin. Brain showed myelin figures due to neuronal degeneration in both the toxicated groups on 30th day. Overall, the results of the study showed that mitochondria was the most affected organelle rendering the pleomorphic (variable sizes and shapes–dumb bell, curved, rounded) mitochondria into opaque structure in the hepatocytes, enterocytes and epithelial cells of kidney tubules. Aflatoxin B1 and ochratoxin treatment in rabbits also revealed similar structural changes reported by [33]. In addition, treatment of ochratoxin (OTA) and citrinin (CTN) to the laboratory rats [61] revealed severe ultrastructural changes in the epithelial cells of PCT characterized by extensive loss of the brush border, cytoplasmic vacuolations and degeneration of nuclei with indistinct nuclear membranes and loss of nucleoli and pleomorphic mitochondria in rats treated with OTA @ 0.75 ppm and CTN @ 15 ppm and in combination. The present study is in agreement with these studies and several others suggesting that toxins indeed has the ability to cause cytotoxic effects in rats, rabbits and poultry [33,62,63].

The mechanism of apoptosis induction is via mitochondrial or non-mitochondrial mechanisms. Although not clear, it is expected to be via triggering the activation of stress-activated kinases c-Jun N-terminal kinase 1 (JNK1) and/or p38 MAPK (Mitogen activated protein kinases) [23,64]. In our study, extensive apoptosis was observed in the higher dose group in all the organs tested. Liver exhibited apoptotic cells in lower dose group also, whereas in the intestinal crypt epithelial cells apoptosis was observed at the higher dose group only. Similarly, the lymphoid and the kidney cells also showed significant apoptosis in a dose-dependent manner.

**Table 9. Effect of T-2 toxin on expression of Heat shock protein (HSP) genes in different tissues in 10ppm and 20ppm groups on 30th day.**

| Tissues | Genes | Groups | Fold change ($2^{-\Delta\Delta ct}$) ±SEM | P value |
|---|---|---|---|---|
| Liver | HSP-72 | 10ppm | 11.93±0.34* | 0.05 |
| | | 20ppm | 21.50±2.22* | |
| | HSP-90 | 10ppm | 4.63±0.79# | 0.05 |
| | | 20ppm | 23.85±4.24# | |
| | HSP-27 | 10ppm | 7.93±0.85$ | 0.05 |
| | | 20ppm | 21.01±0.95$ | |
| Intestine | HSP-72 | 10ppm | 17.19±0.19* | 0.05 |
| | | 20ppm | 38.66±3.94* | |
| | HSP-90 | 10ppm | 14.44±1.69# | 0.05 |
| | | 20ppm | 23.25±2.53# | |
| | HSP-27 | 10ppm | 5.24±0.55$ | 0.05 |
| | | 20ppm | 8.95±0.67$ | |
| MLN | HSP-72 | 10ppm | 4.85±0.53* | 0.05 |
| | | 20ppm | 12.13±1.53* | |
| | HSP-90 | 10ppm | 1.76±0.25# | 0.05 |
| | | 20ppm | 4.98±1.03# | |
| | HSP-27 | 10ppm | 3.91±0.21 | 0.82 |
| | | 20ppm | 3.87±0.9 | |
| kidneys | HSP-72 | 10ppm | 4.13±1.84* | 0.05 |
| | | 20ppm | 11.33±2.2* | |
| | HSP-90 | 10ppm | 4.00±0.64 | 0.51 |
| | | 20ppm | 3.74±0.37 | |
| | HSP-27 | 10ppm | 4.29±0.98 | 0.27 |
| | | 20ppm | 5.51±0.59 | |
| Spleen | HSP-72 | 10ppm | 5.58±0.18* | 0.05 |
| | | 20ppm | 17.71±2.64* | |
| | HSP-90 | 10ppm | 1.35±0.36 | 0.12 |
| | | 20ppm | 1.82±0.59 | |
| | HSP-27 | 10ppm | 5.11±1.51$ | 0.05 |
| | | 20ppm | 3.38±0.69$ | |
| Brain | HSP-72 | 10ppm | 1.7±0.56 | 0.13 |
| | | 20ppm | 4.08±0.79 | |
| | HSP-90 | 10ppm | 2.76±1.84 | 0.13 |
| | | 20ppm | 5.68±1.07 | |
| | HSP-27 | 10ppm | 1.44±0.38 | 0.08 |
| | | 20ppm | 2.62±0.24 | |

The relative fold change ($2^{-\Delta\Delta ct}$) expressions of 10 ppm and 20 ppm groups were compared relative to control on 30th day using Mann–Whitney U-test as implemented in version 16.0 of the SPSS software. The level of significance was set at P < 0.05.

T-2 induced apoptosis has been characterized by some of the previous studies suggesting its effects on the thymic and splenic lymphocytes, bone marrow and gastric mucosa, gastric glandular epithelium and intestinal crypt cell epithelium in mice [23,29,65]particularly in the skin [25]kidney and brain [66].

Crypt epithelial cells of the duodenum and lymphoid tissues have shown extensive apoptosis in previous studies as a result of the toxin treatment to the laboratory mice [65,67,68]. Large

**Table 10. Effect of T-2 toxin on expression of apoptotic genes in different tissues in 10 ppm and 20 ppm groups on 30th day.**

| Tissues | Genes | Groups | Fold change ($2^{-\Delta\Delta ct}$) ±SEM | P value |
|---|---|---|---|---|
| Liver | Caspase 3 | 10ppm | 24.31±1.14* | 0.04 |
| | | 20ppm | 12.15±0.57* | |
| | Bax | 10ppm | 3.85±0.91# | 0.05 |
| | | 20ppm | 9.34±1.52# | |
| | Bcl2 | 10ppm | 0.47±0.11 | 0.17 |
| | | 20ppm | 1.18±0.40 | |
| Intestine | Caspase 3 | 10ppm | 33.52±0.54** | 0.001 |
| | | 20ppm | 67.04±1.08** | |
| | Bax | 10ppm | 12.42±0.45# | 0.05 |
| | | 20ppm | 35.83±4.33# | |
| | Bcl2 | 10ppm | 2.73±0.91$ | 0.05 |
| | | 20ppm | 1.89±0.16$ | |
| MLN | Caspase 3 | 10ppm | 20.87±2.3* | 0.05 |
| | | 20ppm | 41.74±4.6* | |
| | Bax | 10ppm | 43.06±5.52# | 0.05 |
| | | 20ppm | 13.44±2.26# | |
| | Bcl2 | 10ppm | 3.00±0.34$ | 0.05 |
| | | 20ppm | 0.67±0.16$ | |
| kidneys | Caspase 3 | 10ppm | 10.02±0.49* | 0.04 |
| | | 20ppm | 32.78±5.44* | |
| | Bax | 10ppm | 8.82±1.65# | 0.03 |
| | | 20ppm | 4.11±0.83# | |
| | Bcl2 | 10ppm | 3.13±0.24 | 0.57 |
| | | 20ppm | 1.01±0.26 | |
| Spleen | Caspase 3 | 10ppm | 24.97±9.2 | 0.08 |
| | | 20ppm | 15.24±1.15 | |
| | Bax | 10ppm | 21.96±3.56# | 0.05 |
| | | 20ppm | 13.73±0.88# | |
| | Bcl2 | 10ppm | 11.44±3.1$ | 0.05 |
| | | 20ppm | 2.7±0.57$ | |
| Brain | Caspase 3 | 10ppm | 4.27±0.60* | 0.05 |
| | | 20ppm | 1.96±0.20* | |
| | Bax | 10ppm | 1.40±0.43 | 0.5 |
| | | 20ppm | 1.97±0.22 | |
| | Bcl2 | 10ppm | 1.6±0.67 | 0.67 |
| | | 20ppm | 0.31±0.07 | |

The relative fold change ($2^{-\Delta\Delta ct}$) expressions of 10 ppm and 20 ppm groups were compared relative to control on 30th day using Mann–Whitney U-test as implemented in version 16.0 of the SPSS software. The level of significance was set at $P < 0.05$.

intestinal crypts showed relatively lesser apoptosis in comparison to the small intestine probably because the small intestine encounters the toxin and its metabolites more frequently during the mucosal absorption and biliary excretion [10]. With respect to the lymphoid tissues, in a previous study on the laboratory mice it was observed that the degree of apoptosis varied in the different organs but were extensively affect in all the organs studied [68]. Similarly, in the present study the toxin affected all the lymphoid organs at varying degree in a dose dependent manner. Hepatocytes were also significantly affected and graded in our study.

**Table 11. Effect of T-2 toxin on expression of pro-inflammatory cytokine genes in different tissues in 10 ppm and 20 ppm groups on 30[th] day.**

| Tissues | Genes | Groups | Fold change ($2^{-\Delta\Delta ct}$) ±SEM | P value |
|---------|-------|--------|-------------------------------------------|---------|
| Liver | IL-1α | 10ppm | 13.85±2.40* | 0.04 |
| | | 20ppm | 36.37±4.47* | |
| | IL-6 | 10ppm | 3.62±0.18# | 0.05 |
| | | 20ppm | 9.54±0.93# | |
| | TNF-α | 10ppm | 23.85±1.87$ | 0.05 |
| | | 20ppm | 13.29±1.63$ | |
| Intestine | IL-1α | 10ppm | 60.15±4.28* | 0.05 |
| | | 20ppm | 84.25±4.15* | |
| | IL-6 | 10ppm | 11.11±1.01# | 0.04 |
| | | 20ppm | 26.08±4.03# | |
| | TNF-α | 10ppm | 42.12±5.78$ | 0.05 |
| | | 20ppm | 56.13±1.18$ | |
| MLN | IL-1α | 10ppm | 11.72±1.28 | 0.16 |
| | | 20ppm | 7.85±1.10 | |
| | IL-6 | 10ppm | 1.82±0.58 | 0.58 |
| | | 20ppm | 4.22±0.13 | |
| | TNF-α | 10ppm | 13.72±2.19$ | 0.04 |
| | | 20ppm | 6.36±0.46$ | |
| kidneys | IL-1α | 10ppm | 2.14±0.17 | 0.12 |
| | | 20ppm | 4.28±0.48 | |
| | IL-6 | 10ppm | 2.68±0.89# | 0.04 |
| | | 20ppm | 10.95±1.45# | |
| | TNF-α | 10ppm | 19.97±1.13$ | 0.05 |
| | | 20ppm | 8.69±0.88$ | |
| Spleen | IL-1α | 10ppm | 17.72±1.47* | 0.05 |
| | | 20ppm | 25.4±4.76* | |
| | IL-6 | 10ppm | 10.19±0.79## | 0.001 |
| | | 20ppm | 52.23±7.59## | |
| | TNF-α | 10ppm | 49.92±4.30$ | 0.05 |
| | | 20ppm | 12.59±1.22$ | |
| Brain | IL-1α | 10ppm | 1.78±0.50* | 0.05 |
| | | 20ppm | 4.73±0.31* | |
| | IL-6 | 10ppm | 2.42±0.46 | 0.32 |
| | | 20ppm | 3.61±0.28 | |
| | TNF-α | 10ppm | 3.83±1.04 | 0.17 |
| | | 20ppm | 2.17±0.49 | |

The relative fold change ($2^{-\Delta\Delta ct}$) expressions of 10 ppm and 20 ppm groups were compared relative to control on 30[th] day using Mann–Whitney U-test as implemented in version 16.0 of the SPSS software. The level of significance was set at P < 0.05.

Hepatic lesions were graded using the features such as hepatocyte degeneration and necrosis, sinusoidal congestion, centrilobular degeneration and necrosis, hyperplasia, thickening and peri ductular tissue proliferation and fibrosis of the bile duct. A consistent, time- and dose-dependent necrosis was observed in the in the hepatocytes probably due to the exposure to ROS generated by the toxin administration [69]. Similar studies carried out by [70]described hepatic lesions of centro-lobular dystrophy and necrosis of cells of the mononuclear phagocyte

system in rabbits treated with oral dose of 1, 2, 4, 6, 8, 10 or 15 mg/kg body mass of T-2 for 24 to 48 hours.

In addition to the characteristic ultra-structural modifications, toxin treatment also induces activation of heat shock proteins, which are generated by the cells in response to stress, injury and hyperthermia. They are a class of highly conserved stress response proteins expressed ubiquitously in the cells at basal levels under normal conditions. They play the role of molecular chaperons and assist proteins to fold, prevent aggregation of proteins and channelize the unfolded proteins for degradation [71]. They also have a prime role in protecting the cells against oxidative damage by enhancing the protein activity and endogenous antioxidant stability [72].

In the present study, expression of HSP-72,90 and 27 were significantly upregulated in all the organs tested at the toxin concentration of 10 ppm and 20 ppm on 30th day. However, the mRNA expression of hepatic, intestinal and MLN were the highest in group II on the 30th day, which was in line with the study on the mRNA expression in rats using microarray analysis [66]. Kidney tissue homogenate also exhibited significant up regulation of HSP-72 whereas the other HSPs tested (HSP-90 and HSP-27) were not significantly varying. On the contrary, the expression of HSP-72 and HSP-27 were increased in the spleen tissues of the higher dose group whereas that in the brain did not differ remarkably between the two groups but was higher than the control. Previous studies have confirmed the elevated ROS- induced oxidative stress to the cells by the treatment of Zearalenone, citrinin and T-2 toxins [23–25,73]. These studies reveal a strong relationship of oxidative stress and HSPs thus indicating its use as a biomarker to assess oxidative damage to the cells.

Likewise, the expression of pro-apoptotic genes such as Caspase 3 and Bax were also evaluated in the liver, kidney, MLN, intestine, spleen and brain tissues. The expression was significantly higher than the control group in all the organs tested. However, the anti-apoptotic gene (Bcl-2) expression was lower than the control in both the toxin treated groups. Overall, the extensive apoptotic changes induced by T-2 toxin might be due to the significant up regulation of pro-apoptotic genes such as Caspase 3 and Bax as well as reduced expression of anti-apoptotic gene Bcl-2 in different organs. Previous studies have substantiated the present findings suggesting that the toxin treatment regulated the expression of apoptosis-related genes like caspase-2 and insulin-like growth factor binding protein-3, trigger apoptosis by releasing cytochrome *c*, and activate caspase-9, -7, and -3 [66,74,75]. Caspase-3 induces apoptosis by activating a cascade of proteins ultimately resulting in cell death [76].Likewise, the Bcl-2 to Bax ratio is a deciding factor of the fate of cell. Therefore, our study showing significant increase in the Caspase-3 along with a concomitant decrease in the Bcl-2 to Bax ratio suggest the intricate linkage of these two signals in the induction of apoptosis.

Exposure to the trichothecene class of toxins has both immune-stimulatory and suppressive potential based on the dosage, duration and frequency. They target the immune cells such as lymphocytes, macrophages and monocytes leading to the expression of cytokines, chemokines and pro-inflammatory genes [77]. Further, at a greater concentration range they lead to apoptosis of the leukocytes and a corresponding immune suppression. These toxins bind to the ribososmes and activate mitogen-activated protein kinases (MAPKs) along with the regulation of immune and inflammatory genes. Stimulation of mononuclear phagocytes by low doses or concentrations of trichothecenes elicit expression of inflammation-related genes in vivo and in vitro including cyclooxygenase-2 (COX-2), pro-inflammatory cytokines and numerous chemokines [78].

Likewise, the pro-inflammatory cytokines, namely, IL-1α, IL-6 and TNF-α were significantly up in both the groups when compared to the control group on 30th day. It was observed that the expression was more in higher dose exposed animals, which was in agreement with

the earlier studies [78]. Similar studies by [79] reported up regulation of pro-inflammatory cytokines such as IL-1β, IL-6, and TNF-α mRNA in the peritoneal macrophages of mice after exposure to TLR agonists (LPS and DON) and after 16h. Studies have affirmed the up regulation of pro-inflammatory cytokines via the MAPK pathway [80]. Our results uphold these findings because of the significant up regulation of IL-1α and IL-6 mRNA in spleen whereas only IL-1α expression was up regulated in the brain in the group treated with the higher concentration of the toxin. On the contrary, all the organs exhibited a significant increase in the TNF-α gene expression in the lower concentration treated group. TNF-α in responsible for the induction of apoptosis via Fas mediated cascade for the initiation of apoptosis and thus increase in its concentration indicates the activation of this pathway as a result of the toxin treatment.

Infiltration of the lymphocytes and macrophages in the intestinal lamina propria observed in the present study has a strong link to the generation of inflammatory cytokines. This indicates that these cytokines are indeed responsible for the inflammatory response by these cells. Such vascular changes induced chronic anorexia and reduced weight gain, immuno-suppression and tissue injury [81]. Even in the present study, these features were observed in the laboratory mice. Similar findings were elaborated in a recent study on piglets showing DON toxicity caused Salmonella infection, which in turn led to inflammatory responses [82]. The intestinal epithelial cells were made vulnerable for the pathogen attack because of immuno-suppression. Therefore, the lesions observed in our study can be substantiated as a resultant of the inflammation caused by the toxin.

Therefore, the present study establishes a positive association among oxidative stress and the gene markers of stress such as HSPs, the apoptosis inducer genes like Caspase-3 and cytokines by the administration of T-2 toxin. Hence, both oxidative, apoptotic and inflammatory mechanisms are responsible for the pathogenesis of this toxicosis in goats.

## Conclusion

The present study demonstrates a complex relation between HSPs, pro-apoptotic proteins and pro-inflammatory cytokines because of T-2 toxin treatment. This is the first study establishing the effects of T-2 toxin-induced toxicity and pathological changes in juvenile goats. The study shows that T-2 toxin exerts toxic effects through a cumulative effect of oxidative, apoptotic and inflammatory pathways. This study provides a basis for understanding the toxicity, which can be further extrapolated for other animals as well.

## Acknowledgments

The authors are thankful to the Director, Indian Veterinary Research Institute, Izatnagar, India, Director, Central Institute for Research on Goats, Mathura and Joint Director, HSADL, Bhopal, India for providing necessary facilities to carry out the work.

## Author Contributions

**Conceptualization:** Anil Kumar Sharma.

**Data curation:** Rashmi L.

**Formal analysis:** K. Rajukumar.

**Methodology:** Ramith Ramu, Vijay Kumar.

**Project administration:** Anil Kumar Sharma.

**Resources:** Vivek Kumar Gupta.

**Software:** Vijay Kumar.

**Supervision:** Shivasharanappa Nayakwadi, Anil Kumar Sharma.

**Validation:** Vivek Kumar Gupta.

**Visualization:** K. Rajukumar.

**Writing – original draft:** Shivasharanappa Nayakwadi, Ramith Ramu.

**Writing – review & editing:** Shivasharanappa Nayakwadi, Prithvi S. Shirahatti, Rashmi L., Kanthesh M. Basalingappa.

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
