## [Decision Letter · Decision Letter 0]

6 Dec 2019

PONE-D-19-26666

Toxicopathological studies on the effects of T-2 mycotoxin and their interaction in juvenile goats

PLOS ONE

Dear Dr Basalingappa,

Thank you for submitting your manuscript to PLOS ONE. After careful consideration, we feel that it has merit but does not fully meet PLOS ONE’s publication criteria as it currently stands. Therefore, we invite you to submit a revised version of the manuscript that addresses the points raised during the review process.

The reviewers appreciated the work but raised some serious concerns. Please address every single critique raised by both the reviewers for further consideration.

We would appreciate receiving your revised manuscript by Jan 20 2020 11:59PM. To enhance the reproducibility of your results, we recommend that if applicable you deposit your laboratory protocols in protocols.io, where a protocol can be assigned its own identifier (DOI) such that it can be cited independently in the future. For instructions see: http://journals.plos.org/plosone/s/submission-guidelines#loc-laboratory-protocols

We look forward to receiving your revised manuscript.

Kind regards,

Natarajan Aravindan

Academic Editor

PLOS ONE

Journal Requirements:

2) Please amend the manuscript submission data (via Edit Submission) to include authors Anil Kumar Sharma, VivekKumar Gupta., K. Rajukumar, Vijay Kumar and Prithvi S Shirahatti.

3) Please include a copy of Tables 1 to 11 which you refer to in your text.

4) Please include captions for your Supporting Information files at the end of your manuscript, and update any in-text citations to match accordingly. Please see our Supporting Information guidelines for more information: http://journals.plos.org/plosone/s/supporting-information.

Additional Editor Comments (if provided):

Reviewers' comments:

Reviewer's Responses to Questions

**Comments to the Author**

1. Is the manuscript technically sound, and do the data support the conclusions?

Reviewer #1: Yes

Reviewer #2: Yes

2. Has the statistical analysis been performed appropriately and rigorously? 

Reviewer #1: Yes

Reviewer #2: Yes

3. Have the authors made all data underlying the findings in their manuscript fully available?

Reviewer #1: Yes

Reviewer #2: Yes

4. Is the manuscript presented in an intelligible fashion and written in standard English?

Reviewer #1: Yes

Reviewer #2: Yes

5. Review Comments to the Author

Reviewer #1: In this study entitled " Toxicopathological studies on the effects of T-2 mycotoxin and their interaction in juvenile goats", Nayakwadi et al. have delineated the mechanisms by which T-2 toxin induced toxicosis in goats. The authors demonstrated that T-2 toxin elicited significant haematological, biochemical and histological alterations in several tissues including liver, intestine, kidney, spleen, brain etc. They reported altered expression of HSPs, pro-apoptotic and pro-inflammatory cytokines upon exposure to this toxin. Finally, they concluded that the T-2 toxicosis is mediated through oxidative, apoptotic and inflammatory mechanisms in goats. This study is very important as this toxin has been considered to elicit fatal reactions among animals and humans upon consumption. The findings from this study may open a new avenue in understanding the T-2 toxicosis in goats, which may be helpful in diagnosis and prevention of diseases in goats.

The study is well designed and the manuscript is fairly written, however, few queries need to be answered.

Major Comments:

1. Did the authors observe any pathological lesions in rumen or other compartments of stomach during 15th and 30th day of toxin treatment? Any additional data related to this may strengthen the current findings.

2. Under the Materials and Methods, in the lines 169-170, DAPI is mentioned as chromogen. However, the authors have used DAB as chromogen in TUNEL kit. This needs to be corrected. Also the sentences in the lines 183-186 should be removed as the authors have performed real-time RT-PCR and therefore sequencing is not required in mRNA expression studies.

3. Under the results, in line 215, the authors mentioned that "no mortality was observed during the experimental period". Did they observe any abnormal signs in the toxin treated groups during the study? Supportive evidences may be provided in this section.

4. What is the rationale for assessing IL-1apha than IL-1 beta gene expression, since IL-1beta is more potent than IL-1alpha in stimulating IL-6 release in several tissues?

5. The discussion is too descriptive and need to shortened with straightforward discussion.

6. The conclusion needs to be rewritten as it is not clear.

Minor Comments:

1. The authors should mention the name of the authors while referring to the methodology. e.g. Under Materials and Methods, in line 95, the authors mentioned "produced by fermentation of maize and wheat mixture as per the method described by [32]". Similarly, in line 139, the authors mentioned "MLN and brain tissue homogenates for all the animals on 15th and 30th day according to [33]". The authors name should be mentioned in those places.

2. In Figure legend 15: Correct the spelling of " sattellitosis".

3. Cross check the image magnification (both bright field and EM) in all the figures.

4. In the Table 1, under Control, with respect to Route of administration, it has been mentioned "Oral (Mixed with feed)". Did the control feed was mixed with any vehicle?

5. The Table 5 legend reads "T-2 Toxin induced histological alterations (score card) in different organs of young kids on 15th and 30th days fed with 10ppm and 20ppm dose". Remove the word "young kids" and change to "goats".

6. Under "Table 9: Effect of T-2 toxin on expression of heat shock protein (HSP) genes in different tissues in 10ppm and 20ppm groups on 30th day", the P value of some of the HSP mRNA expression levels are greater than P<0.05, which suggest no significant difference between the 10 and 20 ppm toxin treated goats and control. However, the "Expression pattern" suggests "up regulation". So check all the parameters.

7. The authors should strictly follow the references as per the journal format.

8. The manuscript should be checked for grammatical corrections and uniformity to match with the journal format.

Reviewer #2: This study throws light on the effect of T-2 mycotoxin contamination in juvenile goats and holds promise in goat husbandry considering their importance in economy and international trade. The authors have extensively studied the impact of T-2 mycotoxin in goats with respect to their growth and pathophysiology. Following are the queries that the authors need to address before it goes for publication.

1. The authors have stated that such studies have been carried out in other animals like sheep, pigs and poultry. In that case, what is the novelty of this study other than the organism used?

2. It has been mentioned that the toxin was mixed with the feed at specific concentrations. Is there any way the authors could assess the amount of toxin ingested by each animal?

3. The authors could give a possible explanation for the elevated levels of HSP molecules in the brain of control animals as opposed to those treated with the toxin.

4. The figure legends are obscure and very cryptic. It should be more descriptive for the readers to follow the results.

5. There are a few typographical errors that need to be corrected.

6. PLOS authors have the option to publish the peer review history of their article (what does this mean?). If published, this will include your full peer review and any attached files.

Reviewer #1: No

Reviewer #2: No

---

## [Author Response · Author response to Decision Letter 0]

27 Jan 2020

Response to the Reviewer’s comments

Dear Professor

All the authors sincerely thank the editor/s and the reviewers for considering our manuscript significant and providing their valuable time and suggestion to improve the quality of the manuscript. Considering the thoughtful inputs the entire manuscript has been reformed after carrying out necessary incorporations. All the changes incorporated by the authors have been carried using “Track changes” option as suggested. The changes have been followed and remodelled in the respective sections as per the instructions given by the reviewers and as per the guidelines of the journal.

Accordingly, the issues that are raised by reviewers are being addressed in the revised manuscript and here we answer it point-by-point:

Additional requirements Points

1. Please ensure that your manuscript meets PLOS ONE’s style requirements, including those for file naming.

Answer: As suggested the manuscript has been re-checked to meet the PLOS ONE’s style requirements referring to the PLOS ONE journal template.

2) Please amend the manuscript submission data (via Edit Submission) to include authors Anil Kumar Sharma, Vivek Kumar Gupta., K. Rajukumar, Vijay Kumar and Prithvi S Shirahatti.

Answer: As suggested, all the Authors (Anil Kumar Sharma, Vivek Kumar Gupta., K. Rajukumar, Vijay Kumar and Prithvi S Shirahatti) have been included via edit submission.

Also we wish to bring to your kind notice that an author Rashmi L has been included in the manuscript and in edit submission which was not included during the initial stage by mistake and her contribution has also been incorporated in the manuscript.

3) Please include a copy of Tables 1 to 11 which you refer to in your text.

Answer: As suggested, a copy of Tables 1 to 11 referred in the text has been included and submitted.

4) Please include captions for your Supporting Information files at the end of your manuscript, and update any in-text citations to match accordingly.

Answer: The authors would bring to your kind attention that, the Tables 1 to 11 were provided in the supporting information. Now the tables are provided as separate file and the supporting information/supplementary material has been deleted.

Response to the reviewer’s comments

Reviewer 1

The authors sincerely thank the reviewer for the appreciation on the work and their valuable suggestions towards the betterment of the manuscript. 

Major Comments: 

1. Did the authors observe any pathological lesions in rumen or other compartments of stomach during 15th and 30th day of toxin treatment? Any additional data related to this may strengthen the current findings.

Answer: T2 mycotoxin causes extensive pathological lesions in gastro-intestinal system in goats. We observed pathological lesions in rumen and abomasum as well in group II more evidently. There was peeling of ruminal epithelium and diffuse congestion in abomasum mucosa in group II on 15th and 30th day. Histologically, abomasal layer showed severe necrosis and inflammatory cell infiltration in mucosal layer. Group I also showed mild congestion in abomasum on 30th day. There were no significant histological changes in group I and on 15th and 30th day. 

The additional data has been explained in the Results section – clinical signs: Line number 211 to 215. 

2. Under the Materials and Methods, in the lines 169-170, DAPI is mentioned as chromogen. However, the authors have used DAB as chromogen in TUNEL kit. This needs to be corrected. Also the sentences in the lines 183-186 should be removed as the authors have performed real-time RT-PCR and therefore sequencing is not required in mRNA expression studies.

Answer: As suggested, we have used DAB in TUNEL kit and in the lines 169-170 DAPI has been corrected to DAB.

Also, the lines from 183-186 have been removed as per the reviewers suggestion.

3. Under the results, in line 215, the authors mentioned that "no mortality was observed during the experimental period". Did they observe any abnormal signs in the toxin treated groups during the study? Supportive evidences may be provided in this section.

Answer: The experimental trial period was 30 days. Both the toxin groups showed variable degree of clinical signs, which included weakness, lethargy, retardation in growth, disinclination to move and reduced feed intake with feed refusal at 15th day onwards. The most consistent clinical signs were diarrhoea at 20th day onwards in both the groups. In 10 ppm group, 4 animals and in 20 ppm group, all 6 animals showed diarrhoea. Control group (III) kids remained healthy and alert, had normal feed intake and did not exhibit any of these signs throughout the experiment.

The supportive evidences/findings have been discussed & incorporated in the manuscript under the results section – Clinical signs: Line number 204 to 209.

4. What is the rationale for assessing IL-1apha than IL-1 beta gene expression, since IL-1beta is more potent than IL-1alpha in stimulating IL-6 release in several tissues?

Answer: Both IL-1alfa and IL-1beta are involved in inducing reactive oxygen species (ROS) but IL-1 alfa is more potent in causing oxidative damage in the tissues. Also IL-1alfa is strong inducer of TNF-alfa and IL-6 in pro-inflammatory process and causing cell damage. Both the cytokines are able to bind to IL-1R receptor and capable of inducing release of pro-inflammatory cytokines. Therefore, mRNA levels of IL-1alfa were assessed in the T-2 toxin pathogenesis. T-toxin binds to ribosomes and activates mitogen-activated protein kinases (MAPKs) along with the regulation of immune and inflammatory genes and lead to stimulation of mononuclear phagocytes which in turn elicit expression of inflammation-related genes in vivo and in vitro including cyclooxygenase-2 (COX-2), pro-inflammatory cytokines and numerous chemokines. For the activation of macrophages, IL-1alfa is important stimulator; therefore, its level is crucial to assess the immune response and inflammatory response in T-2 toxin.

5. The discussion is too descriptive and need to shorten with straightforward discussion.

Answer: As suggested, the discussion part has been shortened by deleting/modifying which was more descriptive 

6. The conclusion needs to be rewritten as it is not clear.

Answer: As suggested, the whole conclusion has been rewritten for better understanding.

Minor Comments: 

1. The authors should mention the name of the authors while referring to the methodology. e.g. Under Materials and Methods, in line 95, the authors mentioned "produced by fermentation of maize and wheat mixture as per the method described by [32]". Similarly, in line 139, the authors mentioned "MLN and brain tissue homogenates for all the animals on 15th and 30th day according to [33]". The authors name should be mentioned in those places.

Answer: As suggested, in Line 95 & Line 139 of the manuscript the protocol described by AOAC international & Shivasharanappa respectively have been incorporated to refer the methodology. Also the entire manuscript has been checked and followed with the same. 

2. In Figure legend 15: Correct the spelling of "sattellitosis".

Answer: As suggested, the spelling of "sattellitosis" has been changed to "satellitosis".

3. Cross check the image magnification (both bright field and EM) in all the figures.

Answer: As suggested, the image magnification (both bright field and EM) of all the figures have been checked and modified.

4. In the Table 1, under Control, with respect to Route of administration, it has been mentioned "Oral (Mixed with feed)". Did the control feed was mixed with any vehicle?

Answer: As suggested, In the Table 1, under Control, with respect to Route of administration has been changed. The control maintenance ration did not have any vehicle and it was free of any mycotoxins.

5. The Table 5 legend reads "T-2 Toxin induced histological alterations (score card) in different organs of young kids on 15th and 30th days fed with 10ppm and 20ppm dose". Remove the word "young kids" and change to "goats". 

Answer: As suggested, the word "young kids" has been changed to "goats" in Table 5.

6. Under "Table 9: Effect of T-2 toxin on expression of heat shock protein (HSP) genes in different tissues in 10ppm and 20ppm groups on 30th day", the P value of some of the HSP mRNA expression levels are greater than P<0.05, which suggest no significant difference between the 10 and 20 ppm toxin treated goats and control. However, the "Expression pattern" suggests "up regulation". So check all the parameters.

Answer: As suggested, all the parameters have been cross checked and modified. HSP genes expression particularly in brain, spleen and kidney did not differ between toxin treated groups, however there was higher expression then control group. To avoid confusion, the expression pattern in the tables has been discussed in the discussion part. 

7. The authors should strictly follow the references as per the journal format.

Answer: As suggested, the references have formatted as per the journal format. 

8. The manuscript should be checked for grammatical corrections and uniformity to match with the journal format.

Answer: As suggested the entire manuscript has been proofread by a professional expertise for any topological errors and modified accordingly as per the journal format.

Reviewer #2: 

1. The authors have stated that such studies have been carried out in other animals like sheep, pigs and poultry. In that case, what is the novelty of this study other than the organism used?

Answer: In this study, T-2 toxin induced immunopathology and toxicity was studied in goats because, in India goat production is now emerged as big scale enterprise for meat and milk production and intensive animal farming practice which mainly involves feeding of grains, protein feed and other feed resources. Therefore, contamination of such animal feed with mycotoxins is more likely to cause toxicosis in goats. In order to correlate T-2 toxin effects of other animals, the study was conducted in goats. The novelty involved in this study is that, we have observed complex correlation of HSPs, pro-apoptotic proteins and pro-inflammatory cytokines in response to the toxin treatment. These results suggest that the pathogenesis of T-2 toxicosis in goats employs oxidative, apoptotic and inflammatory mechanisms. T-2 toxin induced significant up regulation of HSPs which is novel finding in goats which indicated HSPs play significant role in pathogenesis of T-2 toxicosis.

As suggested, the novelty of the study has also been described in the manuscript in conclusion and introduction part.

2. It has been mentioned that the toxin was mixed with the feed at specific concentrations. Is there any way the authors could assess the amount of toxin ingested by each animal?

Answer: T-2 toxin was grown on maize grains. The culture of T-2 toxin @ 10 and 20 ppm was thoroughly mixed in maintenance ration and aliquots were taken and quantified by TLC and spectrophotometric analysis for assuring the correct concentration. But it was difficult to assess the amount of toxin ingested by each animal. However, daily feeding of maintenance ration (@300g/animal) two times as ensured that every experimental animal was fed as there was no wastage of feed was observed during the trial period.

3. The authors could give a possible explanation for the elevated levels of HSP molecules in the brain of control animals as opposed to those treated with the toxin.

Answer: The expression of HSPs mRNA in brain did not differ between 10 and 20ppm groups but showed upregulation when compared to the control group. 

However, as suggested all the parameters have been cross checked and modified. HSP genes expression particularly in brain, spleen and kidney did not differ between toxin treated groups. To avoid confusion, the expression pattern in the tables has been discussed in the discussion part.

4. The figure legends are obscure and very cryptic. It should be more descriptive for the readers to follow the results.

Answer: As suggested, the figure legends of all the figures have been modified to be more descriptive.

5. There are a few typographical errors that need to be corrected.

Answer: As suggested, the entire manuscript has been checked and proofread for typological errors.

---

## [Editor Report · Decision Letter 1]

7 Feb 2020

Toxicopathological studies on the effects of T-2 mycotoxin and their interaction in juvenile goats

PONE-D-19-26666R1

Dear Dr. Basalingappa,

We are pleased to inform you that your manuscript has been judged scientifically suitable for publication and will be formally accepted for publication once it complies with all outstanding technical requirements.

With kind regards,

Natarajan Aravindan

Academic Editor

PLOS ONE
---

## [Editor Report · Acceptance letter]

28 Feb 2020

PONE-D-19-26666R1 

Toxicopathological studies on the effects of T-2 mycotoxin and their interaction in juvenile goats 

Dear Dr. Basalingappa:

I am pleased to inform you that your manuscript has been deemed suitable for publication in PLOS ONE. Congratulations! Your manuscript is now with our production department. 

With kind regards,

on behalf of

Dr. Natarajan Aravindan 

Academic Editor

PLOS ONE